# RECURSIVE GENERALIZATION TRANSFORMER FOR IMAGE SUPER-RESOLUTION

**Zheng Chen**[1]**, Yulun Zhang**[1][*][†]**, Jinjin Gu**[2,3]**, Linghe Kong**[1][*]**, Xiaokang Yang**[1]
[1]Shanghai Jiao Tong University, [2]Shanghai AI Laboratory, [3]The University of Sydney

## ABSTRACT

Transformer architectures have exhibited remarkable performance in image super-resolution (SR). Since the quadratic computational complexity of the self-attention (SA) in Transformer, existing methods tend to adopt SA in a local region to reduce overheads. However, the local design restricts the global context exploitation, which is crucial for accurate image reconstruction. In this work, we propose the Recursive Generalization Transformer (RGT) for image SR, which can capture global spatial information and is suitable for high-resolution images. Specifically, we propose the recursive-generalization self-attention (RG-SA). It recursively aggregates input features into representative feature maps, and then utilizes cross-attention to extract global information. Meanwhile, the channel dimensions of attention matrices ($query$, $key$, and $value$) are further scaled to mitigate the redundancy in the channel domain. Furthermore, we combine the RG-SA with local self-attention to enhance the exploitation of the global context, and propose the hybrid adaptive integration (HAI) for module integration. The HAI allows the direct and effective fusion between features at different levels (local or global). Extensive experiments demonstrate that our RGT outperforms recent state-of-the-art methods quantitatively and qualitatively. Code and pre-trained models are available at `https://github.com/zhengchen1999/RGT`.

## 1 INTRODUCTION

Image super-resolution (SR) aims to recover a high-resolution (HR) images from its low-resolution (LR) counterpart. Image SR is an ill-posed problem, as there are multiple solutions that can map to any given LR input. To tackle this challenging inverse problem, researchers have proposed numerous deep convolutional neural networks (CNNs) (Dong et al., 2014; Lim et al., 2017; Zhang et al., 2018b; Mei et al., 2020) in the past few years. Thanks to their impressive performance against conventional approaches, CNNs have almost dominated the field of SR.

However, deep CNNs still suffer from a limitation in global context awareness, due to the local processing principle of the convolution operator. Recently, an alternative method, Transformer, has exhibited considerable performance compared with CNN-based methods on multiple high-level computer vision tasks (Dosovitskiy et al., 2021; Liu et al., 2021; Wang et al., 2021; Chu et al., 2021; Yang et al., 2022). Transformer is first developed in the natural language processing (NLP) field, and its core component is the self-attention (SA) mechanism. This mechanism can directly model long-range dependencies by capturing the interaction between all input data. However, the computational complexity of the vanilla self-attention grows quadratically with image size, limiting its application in high-resolution scenarios, especially low-level vision tasks (*e.g.*, image SR).

To apply Transformer in image SR, several methods have been proposed to reduce the computational cost of self-attention. Some researchers apply local (window) self-attention, which divides the feature maps into sub-regions to limit the scope of self-attention. Meanwhile, they utilize the shift mechanism (Liang et al., 2021), overlapping windows (Chen et al., 2022b), or the cross-aggregation operation (Chen et al., 2022c), to enhance the interaction between windows. These methods achieve linear complexity with respect to image size and outperform previous CNN-based methods. However, compared with global attention, the local design needs to stack many blocks to establish global dependencies. Furthermore, some methods propose "transposed" self-attention (Zamir et al., 2022) that operates across the channel dimension instead of the spatial dimension. Although this method

---

[*]Corresponding authors: Yulun Zhang, yulun100@gmail.com; Linghe Kong, linghe.kong@sjtu.edu.cn.
[†]The work was mainly done when Yulun Zhang was at ETH Zürich.

can implicitly capture global information, it hinders modeling spatial dependencies, which is crucial to image SR. Therefore, there is a need to develop a method for image SR that can effectively capture global spatial information with low computational cost on high-resolution images.

In this paper, we propose the Recursive Generalization Transformer (RGT) for image SR, which can model global spatial information and is suitable for high-resolution images. Specifically, we propose the recursive-generalization self-attention (RG-SA) to explore global information directly in linear computational complexity. The RG-SA first generalizes the input features of arbitrary resolution into representative feature maps with a small, constant size, via the recursive generalization module (RGM). Intuitively, the global information is aggregated into representative maps. Then cross-attention is utilized between input features and representative maps to exchange global information. Since the size of representative maps is much smaller than input features, the whole process is at a low computational cost. Moreover, the RG-SA further adjusts the channel dimension of $query$, $key$, and $value$ matrices in SA to mitigate the redundancy in the channel domain.

Furthermore, considering that the RG-SA aggregates the image features via the RGM, it is inevitable to lose some local details. Thus, we combine the RG-SA with the local self-attention (L-SA) in an alternate arrangement to better utilize the global context. To enhance the integration of two different SA modules, we propose the hybrid adaptive integration (HAI), which acts on the outside of each Transformer block. The HAI directly fuses features at different levels (local or global) before and after the block. Besides, HAI adaptively adjusts the input features through a learnable adaptor for feature alignment. Overall, equipped with the above designs, our RGT can capture global information for accurate image SR while the complexity is manageable.

Our contributions can be summarized as follows:

- We propose the Recursive Generalization Transformer (RGT) for image SR. The RGT is capable of capturing global spatial information and is suitable for high-resolution images. Our RGT obtains notable SR performance quantitatively and visually.
- We propose the recursive-generalization self-attention (RG-SA), utilizing the recursive aggregation module and cross-attention to model global dependency with linear complexity.
- We further combine RG-SA with local self-attention to better exploit the global context, and propose the hybrid adaptive integration (HAI) for module integration.

## 2 RELATED WORK

**Image Super-Resolution.** In recent years, CNN-based methods have shown superior performance over conventional SR approaches. SRCNN (Dong et al., 2014) is the pioneering work, that introduces three convolutional layers for image SR. Following this attempt, many works deepen the architecture to improve performance (Zhang et al., 2019; Magid et al., 2021; Dai et al., 2019). VDSR (Kim et al., 2016) introduces residual learning to build a network with 20 layers. EDSR (Lim et al., 2017) further simplifies residual block, which allows deeper networks. RCAN (Zhang et al., 2018a) proposes a residual-in-residual structure to train a model over 400 layers. Moreover, numerous spatial and channel attention mechanisms (Zhang et al., 2019; Liu et al., 2020; Zhou et al., 2020) are proposed to improve the reconstruction quality. For instance, HAN (Niu et al., 2020) proposes a layer attention module and channel-spatial attention. Although these CNN-based methods produce remarkable results, they still suffer from a limitation in global modeling capability.

**Vision Transformer.** Transformer is proposed in natural language processing (NLP) and has been adapted to multiple high-level vision tasks, such as image classification (Dosovitskiy et al., 2021; Liu et al., 2021; Dong et al., 2022), semantic segmentation (Xie et al., 2021), and object detection (Yang et al., 2022; Tu et al., 2022). Due to the impressive performance in high-level tasks, Transformer has also been introduced to low-level vision tasks (Liang et al., 2021; Zamir et al., 2022; Wang et al., 2022; Tsai et al., 2022; Chen et al., 2022c; 2023; Li et al., 2023), including image SR. SwinIR (Liang et al., 2021), following the design of Swin Transformer (Liu et al., 2021), utilizes local window self-attention and shift mechanism. ELAN (Zhang et al., 2022) proposes multi-scale self-attention to reduce the computational cost. CAT (Chen et al., 2022c) designs the rectangle-window self-attention to aggregate the features across different windows. These methods all reduce computational complexity by applying self-attention within local regions. However, the local designs restrict the exploitation of global information that is crucial to image SR.

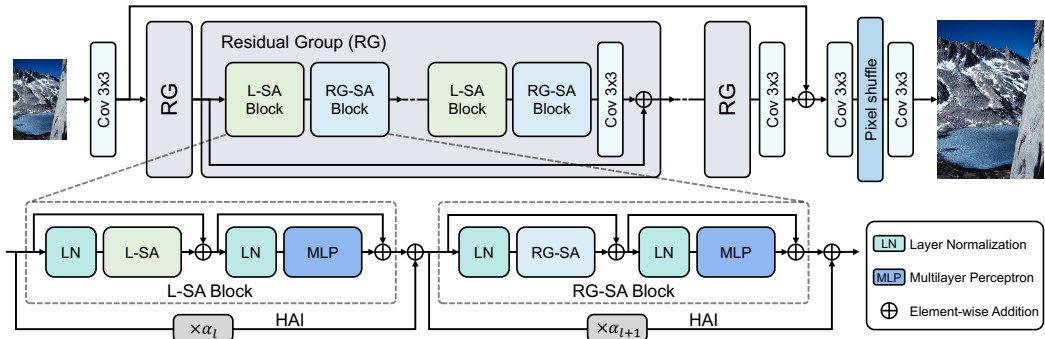

Figure 1: The architecture of the Recursive Generalization Transformer (RGT). The local self-attention (L-SA) blocks, and recursive-generalization self-attention (RG-SA) blocks are alternately arranged. $\alpha_l$ is a learnable adaptor in the hybrid adaptive integration (HAI) of the $l^{th}$ block.

**Global Attention.** To reduce the computational complexity of vanilla self-attention, in addition to local design, several works attempt to propose global attention with low overheads (Wang et al., 2021; Tu et al., 2022; Yang et al., 2022; Chen et al., 2022a; Ali et al., 2021). PVT (Wang et al., 2021) designs a spatial-reduction module to merge tokens of $key$ and $value$. MaxViT (Tu et al., 2022) proposes the grid attention to gain sparse global attention. ScalableViT (Yang et al., 2022) scales attention matrices from both spatial and channel dimensions. Although these methods reduce the computational complexity to a certain extent, the theoretical complexity remains quadratic, which hinders their effective application on high-resolution images. RegionViT Chen et al. (2022a) captures global information among regional tokens to alleviate the overhead of global attention. Moreover, XCiT (Ali et al., 2021) proposes a "transposed" version of self-attention that operates across channels dimension rather than the spatial dimension to achieve linear complexity. However, it cannot explicitly model the spatial relationship. In contrast, we propose the recursive-generalization self-attention (RG-SA), which can explore global spatial information in linear complexity.

## 3 METHOD

We propose the Recursive Generalization Transformer (RGT) for image SR, which is capable of capturing global context and handling high-resolution images effectively. In this section, we first introduce the architecture of RGT. Then, we focus on our proposed recursive-generalization self-attention (RG-SA) and hybrid adaptive integration (HAI).

### 3.1 OVERALL ARCHITECTURE

The overall architecture of our proposed RGT is illustrated in Fig. 1, consisting of three modules: shallow feature extraction, deep feature extraction, and image reconstruction. Given a low-resolution (LR) image $\mathbf{I}_{LR} \in \mathbb{R}^{H \times W \times 3}$, RGT first leverages a convolutional layer as the shallow feature extraction to get the low-level feature $\mathbf{F}_0 \in \mathbb{R}^{H \times W \times C}$, where $H$, $W$, and $C$ represent the image height, width, and channel number. $\mathbf{F}_0$ is used for the deep feature extraction module, which is composed of $N_1$ residual groups (RGs) and a convolutional layer. Through this module, the $\mathbf{F}_0$ is transformed into the deep feature $\mathbf{F}_d \in \mathbb{R}^{H \times W \times C}$. Finally, the $\mathbf{F}_0$ and $\mathbf{F}_d$ are fused through a residual connection, and processed by the reconstruction module to generate the high-resolution (HR) image $\mathbf{I}_{HR} \in \mathbb{R}^{\hat{H} \times \hat{W} \times 3}$, where $\hat{H}$ and $\hat{W}$ are the output height and width. The reconstruction module consists of pixel-shuffle (Shi et al., 2016) and convolutional layers.

Each RG contains $N_2$ Transformer blocks and a convolutional layer. And the residual connection is employed to ensure training stability. As shown in Fig. 1, there are two types of Transformer blocks: Local Self-Attention (L-SA) blocks and RG-SA blocks. The two types of blocks are arranged alternately to organize the topological structure. Each Transformer block is composed of two layer normalization (LN), self-attention, and multilayer perceptron (MLP) (Vaswani et al., 2017). Meanwhile, the HAI acts outside each Transformer block with a learnable adaptor $\alpha$. In this work, we apply the recently proposed rectangle-window self-attention (Rwin-SA) (Chen et al., 2022c) as local self-attention by default. Next, we pay more attention to our proposed RG-SA and HAI.

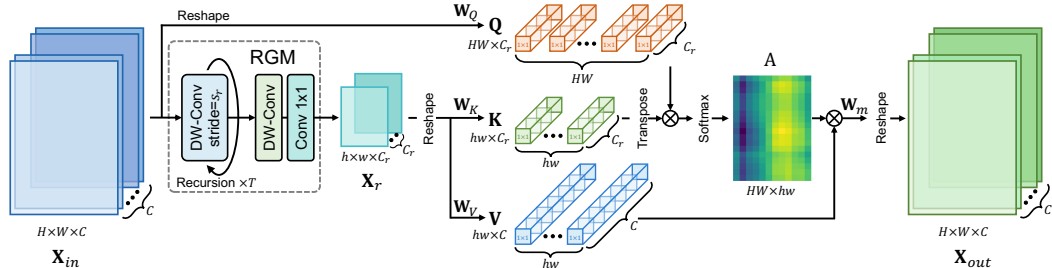

Figure 2: The illustration of the recursive-generalization self-attention (RG-SA). The RG-SA first generates the representative feature maps with constant size ($h \times w$), through the recursive generalization module (RGM). Then, the cross-attention between input features and representative maps is performed to capture global information. $\otimes$ denotes matrix multiplication.

## 3.2 RECURSIVE-GENERALIZATION SELF-ATTENTION

The vanilla self-attention (SA) mechanism establishes connections between all input tokens. Although it can capture global context, the SA suffers the quadratic computational complexity with image size, limiting its the application of SA in high-resolution scenarios, including image SR. To tackle this problem, we propose the recursive-generalization self-attention (RG-SA), shown in Fig. 2, that can maintain linear computational complexity while capturing global information.

The RG-SA first aggregates input image features of arbitrary resolution into compressed maps (denoted as representative feature maps), through the recursive generalization module (RGM). Intuitively, the representative maps aggregate the information of the whole image features, providing a global view of the image. Then, the cross-attention is calculated between input features and representative maps. Therefore, each token in the input image features can obtain a global receptive field. Meanwhile, we further scale the channel dimension of $query$, $key$, and $value$ matrices in attention to mitigate the channel redundancy. It improves the performance and reduces consumption.

**Recursive Generalization Module.** For simplicity, our RGM only consists of depth-wise and pixel-wise convolutions, shown in Fig. 2. Given an input $\mathbf{X}_{in} \in \mathbb{R}^{H \times W \times C}$, we first compress spatial size of the features by recursively reusing a single depth-wise convolution $T = \lfloor \log_{s_r} \frac{H}{h} \rfloor$ times to obtain the rough aggregation maps $\hat{\mathbf{X}} \in \mathbb{R}^{h \times w \times C}$, where $s_r$ is the convolution stride size, $h$ is a constant, and $w = W \times \frac{h}{H}$. Without loss of generality, we assume $W \le H$, then $w \le h$. Next, we refine the rough aggregation maps to generate the representative maps $\mathbf{X}_r \in \mathbb{R}^{h \times w \times C_r}$, through a $3 \times 3$ depth-wise convolution and a $1 \times 1$ point-wise convolution. The RGM is formulated as:

$$\hat{\mathbf{X}} = \mathbf{W}_r^K(\mathbf{X}_{in}) = \mathbf{W}_r(\mathbf{W}_r(\dots(\mathbf{W}_r(\mathbf{X}_{in})))),$$
$$\mathbf{X}_r = \mathbf{W}_p \mathbf{W}_d(\hat{\mathbf{X}}), \tag{1}$$

where $\mathbf{W}_r$ is the depth-wise convolution with $s_r$ stride, $\mathbf{W}_d$ is the $3 \times 3$ depth-wise convolution, and $\mathbf{W}_p$ is the $1 \times 1$ point-wise convolution. Also, the $1 \times 1$ point-wise convolution scales the channels from $C$ to $C_r = C \times c_r$, where $c_r$ is the adjustment factor. Through RGM, we can aggregate the global information of the input image features. Meanwhile, the recursive design is flexible for processing inputs with varying sizes (common in image SR) by dynamically choosing the recursion times $T$.

**Cross-Attention.** Subsequently, we reshape and project the input features $\mathbf{X}_{in}$ as the $\mathbf{Q} \in \mathbb{R}^{HW \times C_r}$ ($query$) and the representative maps as the $\mathbf{K} \in \mathbb{R}^{hw \times C_r}$ ($key$), and $\mathbf{V} \in \mathbb{R}^{hw \times C}$ ($value$) to compute the cross-attention. The attention matrix $\mathbf{A} \in \mathbb{R}^{HW \times hw}$ is calculated from the dot-product interaction of $query$ and $key$. Note that we further scale the channel dimensions of $query$, $key$, and $value$. Overall, the whole cross-attention process is defined as:

$$\mathbf{Q} = \mathbf{W}_Q \mathbf{X}_{in}, \mathbf{K} = \mathbf{W}_K \mathbf{X}_r, \mathbf{V} = \mathbf{W}_V \mathbf{X}_r,$$
$$\mathbf{A} = \text{SoftMax}(\mathbf{Q}\mathbf{K}^T / \sqrt{C_r}), \tag{2}$$
$$\text{Cross-Attention}(\mathbf{X}_{in}, \mathbf{X}_r) = \mathbf{W}_m(\mathbf{A} \cdot \mathbf{V}),$$

where $\mathbf{W}_Q \in \mathbb{R}^{C \times C_r}$, $\mathbf{W}_K \in \mathbb{R}^{C_r \times C_r}$, and $\mathbf{W}_V \in \mathbb{R}^{C_r \times C}$ are learnable parameters and biases are omitted for simplification; $\mathbf{W}_m \in \mathbb{R}^{C \times C}$ is the projection matrix for feature fusion. Similar to vanilla self-attention (Vaswani et al., 2017; Dosovitskiy et al., 2021), we divide the channels into

multiple "heads" and execute the attention operation in parallel. Finally, we reshape the result of cross-attention to obtain the output features $\mathbf{X}_{out}\in\mathbb{R}^{H\times W\times C}$. Through RGM, cross-attention, and channel adjustment, our RG-SA can capture global spatial information while maintaining low computational overheads. Next, we analyze the complexity of the RG-SA in detail.

**Complexity Analysis.** Our RG-SA can be divided into two components: RGM and cross-attention. For RGM, the computational complexity is $\mathcal{O}(HWC)$. For cross-attention, the computational complexity is $\mathcal{O}(hwHW(C+C_r)+HWC(C+C_r)+hwC_r(C+C_r))$. Here, we analyze the single-head self-attention for simplicity. Since $h$ and $w$ are constants, and $C_r=C\times c_r$, the complexity of cross-attention is $\mathcal{O}(HWC^2)$. In general, The total computational complexity of our RG-SA is linear with the input features size ($H\times W$). Additionally, by applying the small ($<1$) adjustment factor $c_r$, we can mitigate the channel redundancy, thus further reducing complexity.

### 3.3 Hybrid Adaptive Integration

**Alternate Arrangement.** In RG-SA, the global information is captured through the cross-attention between input features and representative maps, ensuring low computational overheads. However, the RGM in RG-SA is a coarse-grained design, which leads to losing some local details and ultimately limits modeling global information. To improve the exploitation of the global context, we introduce local self-attention (L-SA) and combine it with our proposed RG-SA. Two attention modules are alternately arranged in each residual group (RG), as illustrated in Fig. 1.

**Analysis and Motivation.** Although the two blocks are integrated, the linear arrangement lacks direct interaction between features at different levels (global or local), thus still cannot exploit global information effectively. For further analysis, under the alternating topology, the input and output of each Transformer block are different level features. Specifically, the input of the RG-SA block is the local features generated from L-SA, while the output is the global features. Correspondingly, the input and output of L-SA are global and local features, respectively. This observation inspires us to enhance information fusion by combining the input and output features of each block.

**Specific Design.** The intuitive idea is directly integrating the input and output features via the vanilla skip connection (He et al., 2016). Nevertheless, since the misalignment between global and local features, simple addition cannot fuse features effectively. To overcome the above issues, we propose hybrid adaptive integration (HAI). As shown in Fig. 1, our HAI acts on the outside of each Transformer block. The input features are adaptively adjusted by a learnable adaptor $\alpha$, and added to the output features. The process of the $l^{th}$ Transformer block $\mathcal{B}^l$ equipped with HAI is:

$$Z^l = \mathcal{B}^l(Z^{l-1}) + \alpha_l \cdot Z^{l-1}, \tag{3}$$

where $Z^{l-1}$ and $Z^l$ represent the input and the output of the $l^{th}$ Transformer block, $\alpha_l\in\mathbb{R}^C$ is the learnable parameter in the $l^{th}$ block. Overall, the HAI is able to enhance the integration of different SA modules on the basis of the alternate arrangement, which advances the modeling of global information. Moreover, similar to the regular residual connection, our HIA encourages more information flows to the deep network layers, resulting in better performance.

**Visual Results.** To intuitively show the effectiveness of HAI, we visualize the relevant features of one RG-SA block (first row, $2^{nd}$ block in $1^{st}$ RG) and one L-SA block (second row, $5^{td}$ block in $1^{st}$ RG) of RGT in Fig. 3. The deeper color indicates larger weights. First, by observing columns (a) and (b), we can find great differences in the input and output of SA modules in some cases. It reveals the misalignment between global and local features. Secondly, directly fused by the vanilla skip connection, the features are

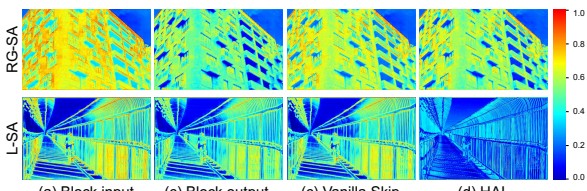

(a) Block input  (c) Block output  (c) Vanilla Skip  (d) HAI

Figure 3: Visualization of the features in RGT. a) Block input. (b) Block output before connection. (c) Vanilla skip connection result: directly add input to output. (d) HAI result: adjust input with $\alpha$ before adding to output. Please zoom in for a better visualization.

over-integrated (top of (c) column), or not changed obviously (bottom of (c) column). In contrast, through HAI, the features shown in column (d) are adaptively changed according to different blocks. It indicates that the input and output features are fused effectively. More discussions on the role of HAI and the value distribution of learnable adaptors $\alpha$ are given in Sec. 4.2.

| L-SA | RG-SA | HAI | Params(M) | FLOPs(G) | PSNR(dB) | SSIM |
|------|-------|-----|-----------|----------|----------|------|
| ✓ | | | 10.69 | 229.42 | 33.43 | 0.9396 |
| ✓ | ✓ | | 10.04 | 183.08 | 33.52 | 0.9405 |
| ✓ | ✓ | ✓ | 10.05 | 183.08 | 33.68 | 0.9414 |

(a) Break-down ablation on each component.

| Method | Recursion | $c_r$ | Params(M) | FLOPs(G) | PSNR(dB) | SSIM |
|--------|-----------|-------|-----------|----------|----------|------|
| w/o Recur | | 0.5 | 10.05 | 274.54 | 33.57 | 0.9412 |
| w/o Scale | ✓ | 1 | 11.37 | 189.62 | 33.54 | 0.9404 |
| RGT-S | ✓ | 0.5 | 10.05 | 183.08 | 33.68 | 0.9414 |

(b) Ablation study on RG-SA.

| Method | Vanilla Skip | $\alpha$ | Params (M) | FLOPs (G) | PSNR (dB) | SSIM |
|--------|--------------|----------|------------|-----------|-----------|------|
| w/o HAI | | | 10.04 | 183.08 | 33.52 | 0.9405 |
| w/ Skip | ✓ | | 10.04 | 183.08 | 32.71 | 0.9339 |
| w/ HAI | ✓ | ✓ | 10.05 | 183.08 | 33.68 | 0.9414 |

(c) Ablation study on HAI.

| Method | Params(M) | FLOPs(G) | PSNR(dB) | SSIM |
|--------|-----------|----------|----------|------|
| L-SA only | 10.69 | 229.42 | 33.43 | 0.9396 |
| L-SA w/ HAI | 10.69 | 229.42 | 33.44 | 0.9400 |

(d) Further ablation study on HAI.

Table 1: Ablation studies. We train models on DIV2K and Flickr2K, and test on Urban100 ($\times 2$).

## 4 EXPERIMENTS

### 4.1 EXPERIMENTAL SETTINGS

**Data and Evaluation.** Following recent works (Zhang et al., 2021; Magid et al., 2021; Liang et al., 2021), we choose DIV2K (Timofte et al., 2017) and Flickr2K (Lim et al., 2017) as the training data. For testing, we use five standard benchmark datasets: Set5 (Bevilacqua et al., 2012), Set14 (Zeyde et al., 2010), B100 (Martin et al., 2001), Urban100 (Huang et al., 2015), and Manga109 (Matsui et al., 2017). We conduct experiments with three upscaling factors: $\times 2$, $\times 3$, and $\times 4$. The low-resolution images are generated by Bicubic (BI) downsampling. To evaluate our method, we use the metrics PSNR and SSIM (Wang et al., 2004) on the Y channel of the YCbCr space.

**Implementation Details.** We set two version models, RGT-S and RGT, with different computational complexity. For RGT-S, we set the residual group (RG) number $N_1$ as 6 and the Transformer block number $N_2$ for each RG as 6 for L-SA. The channel dimension number, attention head number, and mlp expansion ratio (Dosovitskiy et al., 2021) are set as 180, 6, and 2, respectively. The window size is set as $8 \times 32$. The $s_r$ (stride size) and $c_r$ (adjustment factor) are set as 4 and 0.5 for RG-SA. The representative map size $h$ is set as 4 for training, and 16 for testing. For RGT, we increase the number of RG from 6 to 8, while other settings remain the same as RGT-S.

**Training Settings.** We train our models with batch size 32, where each input image is randomly cropped to $64 \times 64$ size, and the total training iterations are 500K. Training patches are augmented using random horizontal flips and rotations with $90°$, $180°$, and $270°$. To keep fair comparisons, we adopt Adam optimizer (Kingma & Ba, 2015) with $\beta_1 = 0.9$ and $\beta_2 = 0.99$ to minimize the $L_1$ loss function following previous works (Zhang et al., 2018a; Dai et al., 2019; Liang et al., 2021). The initial learning rate is set as $2 \times 10^{-4}$ and reduced by half at the milestone [250K,400K,450K,475K]. We use PyTorch (Paszke et al., 2019) to implement our models with 4 Nvidia A100 GPUs.

### 4.2 ABLATION STUDY

In this section, we study the effects of different components of our method. For fair comparisons, all models adopt the same basic architecture and settings as RGT-S. We conduct experiments on the $\times 2$ factor. We adopt the dataset DIV2K (Timofte et al., 2017) and Flickr2K (Lim et al., 2017) to train models, and the iterations are 200K. The dataset Urban100 (Huang et al., 2015) is applied for testing. When we calculate the FLOPs, the input size is $3 \times 128 \times 128$.

**Effects of each component.** We conduct a break-down ablation experiment to investigate the effects of each component on SR performance. The results are listed in Tab. 1a. *First, baseline.* The baseline model is derived by replacing all Transformer blocks in RGT-S with local self-attention (L-SA) (Liang et al., 2021; Chen et al., 2022b) block and removing HAI. We set the window size of L-SA as $8 \times 32$, which is consistent with RGT-S. *Second, applying RG-SA.* We introduce the recursive-generalization self-attention (RG-SA) into the baseline, and alternately arrange L-SA and RG-SA in successive Transformer blocks. Without changing the structure of the network, the model achieves a 0.09 dB improvement. Meanwhile, compared with the baseline, the FLOPs and parameters of the model are slightly reduced. It shows that our proposed RG-SA is effective regarding parameters and computational complexity. *Third, applying HAI.* We further adopt the hybrid adaptive integration (HAI) and get the final version of our RGT-S. The model obtains the best performance of 33.68 dB. These results demonstrate the effectiveness of the RG-SA and HAI.

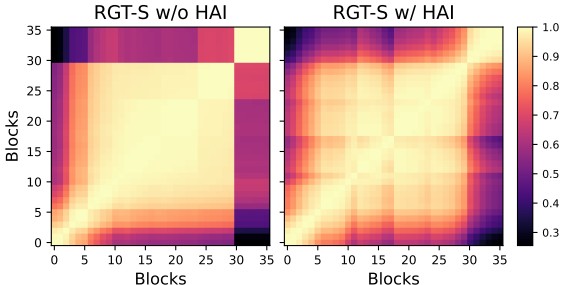 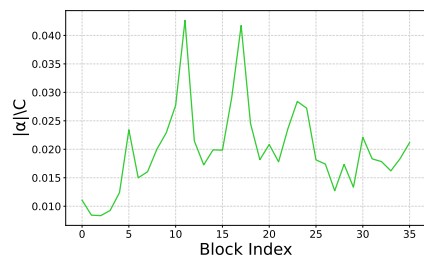

Figure 4: CKA similarity between all pairs of Transformer blocks in RGT-S without or with HAI.

Figure 5: The value ($|\alpha|/C$, $C$=180) distribution of $\alpha$ of RGT-S.

**Effects of RG-SA.** We investigate the design of our recursive-generalization self-attention (RG-SA). We implement ablation experiments on the recursive operation and channel adjustment factor. The results are reported in Tab. 1b. ***First, the impact of recursive operation.*** We build the model without recursion (w/o Recur) by removing the recursive operation of RGM in RGT-S. Namely, the stride depth-wise convolution is only utilized once to generate the representative maps. We keep $s_r$=4 and $c_r$=0.5 unchanged. Compared with RGT-S, it can be observed that the usage of recursive operation can effectively reduce the FLOPs by 30%, while achieving better performance. ***Second, the impact of the channel adjustment factor $c_r$.*** We build and train the model without channel scaling (w/o Scale) by setting the adjustment factor $c_r$ in RGT-S as 1 (0.5 by default). Compared with RGT-S, we discover that scaling the channel dimension yields 0.14 dB performance gain, since the smaller $c_r$ mitigates the redundancy between channels, thus enhancing the feature expression.

**Effects of HAI.** We show the influence of the hybrid adaptive integration (HAI) in Tab. 1c. We compared three models: without HAI (w/o HAI), with vanilla skip connection (He et al., 2016) (w/ Skip), and with HAI (w/ HAI). Both skip connection and HAI act on the outside of each Transformer block. ***First, the impact of vanilla skip connection.*** Comparing the model w/o HAI and the model w/ Skip, we can find that the simple application of skip connection seriously degrades the model performance by 0.81 dB. This may be due to the misalignment between different level (global or local) features, which prevents their direct fusion. ***Second, the impact of HAI.*** In contrast, our proposed HAI adaptively adjusts the input features through a learnable adaptor $\alpha$, thus achieving valid feature integration. With the HAI, the model obtains 0.16 dB gain.

We also apply HAI only models only with L-SA in Tab. 1d. Since there is no feature misalignment, the performance does not change much with HAI. These results are consistent with analysis in Sec. 3.3 and demonstrates the effectiveness of HAI in solving feature misalignment.

Furthermore, we further introduce centered kernel alignment (CKA) (Cortes et al., 2012; Kornblith et al., 2019; Raghu et al., 2021) to study the internal representation structure of the model. The higher the CKA score, the higher the representation similarity. We calculate CKA similarities between all Transformer blocks in RGT-S without and with HAI. The results are shown as heatmaps in Fig. 4. ***First, enhance module integration.*** There are obvious differences before and after the $30^{th}$ block, in the model without HAI. On the contrary, equipping with HAI, it can be found that the transition is more gradual, which indicates that the integration between the modules is more effective. ***Second, encourage information flow.*** We observe that there is still a high similarity between the initial blocks ($0^{th}$~$2^{nd}$) and the very deep blocks ($28^{th}$~$29^{th}$) in the model with HAI. It means that through HAI, more information can flow to the deep layers of the network.

We also visualize the value distribution of $\alpha$ of RGT-S in Fig. 5. We find that $\alpha$ is diverse based on different blocks, indicating its adaptability. Meanwhile, the value of $\alpha$ increases at the end block in each RG, *i.e.*, blocks $5^{th}$, $11^{th}$, $17^{th}$, and $23^{th}$. It indicates that HAI integrates different modules.

### 4.3 COMPARISONS WITH STATE-OF-THE-ART METHODS

We compare our two models, RGT-S and RGT, with recent state-of-the-art methods: EDSR (Lim et al., 2017), RCAN (Zhang et al., 2018a), SRFBN (Li et al., 2019), SAN (Dai et al., 2019), HAN (Niu et al., 2020), CSNLN (Mei et al., 2020), NLSA (Mei et al., 2021), CRAN (Zhang et al., 2021), DFSA (Magid et al., 2021), ELAN (Zhang et al., 2022), SwinIR (Liang et al., 2021), and CAT-A (Chen et al., 2022c). Similar to previous works (Lim et al., 2017; Zhang et al., 2018a), we use self-ensemble strategy in testing and mark the model with the symbol "+".

| Method | Scale | Set5 | | Set14 | | B100 | | Urban100 | | Manga109 | |
|---|---|---|---|---|---|---|---|---|---|---|---|
| | | PSNR | SSIM | PSNR | SSIM | PSNR | SSIM | PSNR | SSIM | PSNR | SSIM |
| EDSR (Lim et al., 2017) | ×2 | 38.11 | 0.9602 | 33.92 | 0.9195 | 32.32 | 0.9013 | 32.93 | 0.9351 | 39.10 | 0.9773 |
| RCAN (Zhang et al., 2018a) | ×2 | 38.27 | 0.9614 | 34.12 | 0.9216 | 32.41 | 0.9027 | 33.34 | 0.9384 | 39.44 | 0.9786 |
| SRFBN (Li et al., 2019) | ×2 | 38.11 | 0.9609 | 33.82 | 0.9196 | 32.29 | 0.9010 | 32.62 | 0.9328 | 39.08 | 0.9779 |
| SAN (Dai et al., 2019) | ×2 | 38.31 | 0.9620 | 34.07 | 0.9213 | 32.42 | 0.9028 | 33.10 | 0.9370 | 39.32 | 0.9792 |
| HAN (Niu et al., 2020) | ×2 | 38.27 | 0.9614 | 34.16 | 0.9217 | 32.41 | 0.9027 | 33.35 | 0.9385 | 39.46 | 0.9785 |
| CSNLN (Mei et al., 2020) | ×2 | 38.28 | 0.9616 | 34.12 | 0.9223 | 32.40 | 0.9024 | 33.25 | 0.9386 | 39.37 | 0.9785 |
| NLSA (Mei et al., 2021) | ×2 | 38.34 | 0.9618 | 34.08 | 0.9231 | 32.43 | 0.9027 | 33.42 | 0.9394 | 39.59 | 0.9789 |
| CRAN (Zhang et al., 2021) | ×2 | 38.31 | 0.9617 | 34.22 | 0.9232 | 32.44 | 0.9029 | 33.43 | 0.9394 | 39.75 | 0.9793 |
| DFSA (Magid et al., 2021) | ×2 | 38.38 | 0.9620 | 34.33 | 0.9232 | 32.50 | 0.9036 | 33.66 | 0.9412 | 39.98 | 0.9798 |
| ELAN (Zhang et al., 2022) | ×2 | 38.36 | 0.9620 | 34.20 | 0.9228 | 32.45 | 0.9030 | 33.44 | 0.9391 | 39.62 | 0.9793 |
| SwinIR (Liang et al., 2021) | ×2 | 38.42 | 0.9623 | 34.46 | 0.9250 | 32.53 | 0.9041 | 33.81 | 0.9427 | 39.92 | 0.9797 |
| CAT-A (Chen et al., 2022c) | ×2 | 38.51 | 0.9626 | 34.78 | 0.9265 | 32.59 | 0.9047 | 34.26 | 0.9440 | 40.10 | 0.9805 |
| RGT-S (ours) | ×2 | 38.56 | 0.9627 | 34.77 | 0.9270 | 32.59 | 0.9050 | 34.32 | 0.9457 | 40.18 | 0.9805 |
| RGT (ours) | ×2 | 38.59 | 0.9628 | 34.83 | 0.9272 | 32.62 | 0.9050 | 34.47 | 0.9467 | 40.34 | 0.9808 |
| RGT+ (ours) | ×2 | 38.62 | 0.9629 | 34.88 | 0.9275 | 32.64 | 0.9053 | 34.63 | 0.9474 | 40.45 | 0.9810 |
| EDSR (Lim et al., 2017) | ×3 | 34.65 | 0.9280 | 30.52 | 0.8462 | 29.25 | 0.8093 | 28.80 | 0.8653 | 34.17 | 0.9476 |
| RCAN (Zhang et al., 2018a) | ×3 | 34.74 | 0.9299 | 30.65 | 0.8482 | 29.32 | 0.8111 | 29.09 | 0.8702 | 34.44 | 0.9499 |
| SRFBN (Li et al., 2019) | ×3 | 34.70 | 0.9292 | 30.51 | 0.8461 | 29.24 | 0.8084 | 28.73 | 0.8641 | 34.18 | 0.9481 |
| SAN (Dai et al., 2019) | ×3 | 34.75 | 0.9300 | 30.59 | 0.8476 | 29.33 | 0.8112 | 28.93 | 0.8671 | 34.30 | 0.9494 |
| HAN (Niu et al., 2020) | ×3 | 34.75 | 0.9299 | 30.67 | 0.8483 | 29.32 | 0.8110 | 29.10 | 0.8705 | 34.48 | 0.9500 |
| CSNLN (Mei et al., 2020) | ×3 | 34.74 | 0.9300 | 30.66 | 0.8482 | 29.33 | 0.8105 | 29.13 | 0.8712 | 34.45 | 0.9502 |
| NLSA (Mei et al., 2021) | ×3 | 34.85 | 0.9306 | 30.70 | 0.8485 | 29.34 | 0.8117 | 29.25 | 0.8726 | 34.57 | 0.9508 |
| CRAN (Zhang et al., 2021) | ×3 | 34.80 | 0.9304 | 30.73 | 0.8498 | 29.38 | 0.8124 | 29.33 | 0.8745 | 34.84 | 0.9515 |
| DFSA (Magid et al., 2021) | ×3 | 34.92 | 0.9312 | 30.83 | 0.8507 | 29.42 | 0.8128 | 29.44 | 0.8761 | 35.07 | 0.9525 |
| ELAN (Zhang et al., 2022) | ×3 | 34.90 | 0.9313 | 30.80 | 0.8504 | 29.38 | 0.8124 | 29.32 | 0.8745 | 34.73 | 0.9517 |
| SwinIR (Liang et al., 2021) | ×3 | 34.97 | 0.9318 | 30.93 | 0.8534 | 29.46 | 0.8145 | 29.75 | 0.8826 | 35.12 | 0.9537 |
| CAT-A (Chen et al., 2022c) | ×3 | 35.06 | 0.9326 | 31.04 | 0.8538 | 29.52 | 0.8160 | 30.12 | 0.8862 | 35.38 | 0.9546 |
| RGT-S (ours) | ×3 | 35.11 | 0.9328 | 31.05 | 0.8548 | 29.53 | 0.8164 | 30.18 | 0.8884 | 35.39 | 0.9548 |
| RGT (ours) | ×3 | 35.15 | 0.9329 | 31.13 | 0.8550 | 29.55 | 0.8165 | 30.28 | 0.8899 | 35.55 | 0.9553 |
| RGT+ (ours) | ×3 | 35.18 | 0.9331 | 31.16 | 0.8558 | 29.57 | 0.8170 | 30.40 | 0.8914 | 35.69 | 0.9559 |
| EDSR (Lim et al., 2017) | ×4 | 32.46 | 0.8968 | 28.80 | 0.7876 | 27.71 | 0.7420 | 26.64 | 0.8033 | 31.02 | 0.9148 |
| RCAN (Zhang et al., 2018a) | ×4 | 32.63 | 0.9002 | 28.87 | 0.7889 | 27.77 | 0.7436 | 26.82 | 0.8087 | 31.22 | 0.9173 |
| SRFBN (Li et al., 2019) | ×4 | 32.47 | 0.8983 | 28.81 | 0.7868 | 27.72 | 0.7409 | 26.60 | 0.8015 | 31.15 | 0.9160 |
| SAN (Dai et al., 2019) | ×4 | 32.64 | 0.9003 | 28.92 | 0.7888 | 27.78 | 0.7436 | 26.79 | 0.8068 | 31.18 | 0.9169 |
| HAN (Niu et al., 2020) | ×4 | 32.64 | 0.9002 | 28.90 | 0.7890 | 27.80 | 0.7442 | 26.85 | 0.8094 | 31.42 | 0.9177 |
| CSNLN (Mei et al., 2020) | ×4 | 32.68 | 0.9004 | 28.95 | 0.7888 | 27.80 | 0.7439 | 27.22 | 0.8168 | 31.43 | 0.9201 |
| NLSA (Mei et al., 2021) | ×4 | 32.59 | 0.9000 | 28.87 | 0.7891 | 27.78 | 0.7444 | 26.96 | 0.8109 | 31.27 | 0.9184 |
| CRAN (Zhang et al., 2021) | ×4 | 32.72 | 0.9012 | 29.01 | 0.7918 | 27.86 | 0.7460 | 27.13 | 0.8167 | 31.75 | 0.9219 |
| DFSA (Magid et al., 2021) | ×4 | 32.79 | 0.9019 | 29.06 | 0.7922 | 27.87 | 0.7458 | 27.17 | 0.8163 | 31.88 | 0.9266 |
| ELAN (Zhang et al., 2022) | ×4 | 32.75 | 0.9022 | 28.96 | 0.7914 | 27.83 | 0.7459 | 27.13 | 0.8167 | 31.68 | 0.9226 |
| SwinIR (Liang et al., 2021) | ×4 | 32.92 | 0.9044 | 29.09 | 0.7950 | 27.92 | 0.7489 | 27.45 | 0.8254 | 32.03 | 0.9260 |
| CAT-A (Chen et al., 2022c) | ×4 | 33.08 | 0.9052 | 29.18 | 0.7960 | 27.99 | 0.7510 | 27.89 | 0.8339 | 32.39 | 0.9285 |
| RGT-S (ours) | ×4 | 32.98 | 0.9047 | 29.18 | 0.7966 | 27.98 | 0.7509 | 27.89 | 0.8347 | 32.38 | 0.9281 |
| RGT (ours) | ×4 | 33.12 | 0.9060 | 29.23 | 0.7972 | 28.00 | 0.7513 | 27.98 | 0.8369 | 32.50 | 0.9291 |
| RGT+ (ours) | ×4 | 33.16 | 0.9066 | 29.28 | 0.7979 | 28.03 | 0.7520 | 28.09 | 0.8388 | 32.68 | 0.9303 |

Table 2: Quantitative comparison (PSNR/SSIM) with state-of-the-art methods. Best and second best results are colored with red and blue. Our methods outperforms other competitors.

**Quantitative results.** We show the quantitative comparisons for ×2, ×3, and ×4 image SR in Tab. 2. We also report the comparisons of computational complexity (*e.g.*, FLOPs), and parameter numbers in Tab. 3. As we can see, our proposed RGT significantly outperforms other methods on all datasets with all scaling factors. Compared with recent Transformer-based methods, such as SwinIR (Liang et al., 2021) and CAT-A (Chen et al., 2022c), our proposed RGT achieves better results, particularly on the Urban100 and Manga109 datasets. For instance, on the Urban100 dataset (×2), RGT outperforms CAT-A by 0.21 dB. Meanwhile, the model size and computational complexity are lower than CAT-A. Even the small vision model, RGT-S, obtains comparable or better results than compared methods. These comparisons indicate that our proposed RGT can capture more global information compared with previous CNN-based and Transformer-based methods.

**Visual Results.** We show visual comparisons (×4) in Fig. 6. We can observe that most compared methods suffer from blurring artifacts and cannot recover accurate textures in some representative challenging cases. In contrast, our RGT can alleviate the blurring artifacts better and recover more image details. For instance, in image img_059, some methods fail to reconstruct most of the strips correctly (*e.g.*, SAN and DFSA), while some only restore part stripes (*e.g.*, SwinIR and CAT-A). In contrast, our method recovers more precise structures. These visual comparisons demonstrate that our RGT is capable of reconstructing high-quality images by modeling global information. Combining with the quantitative comparisons, we further demonstrate the effectiveness of our method.

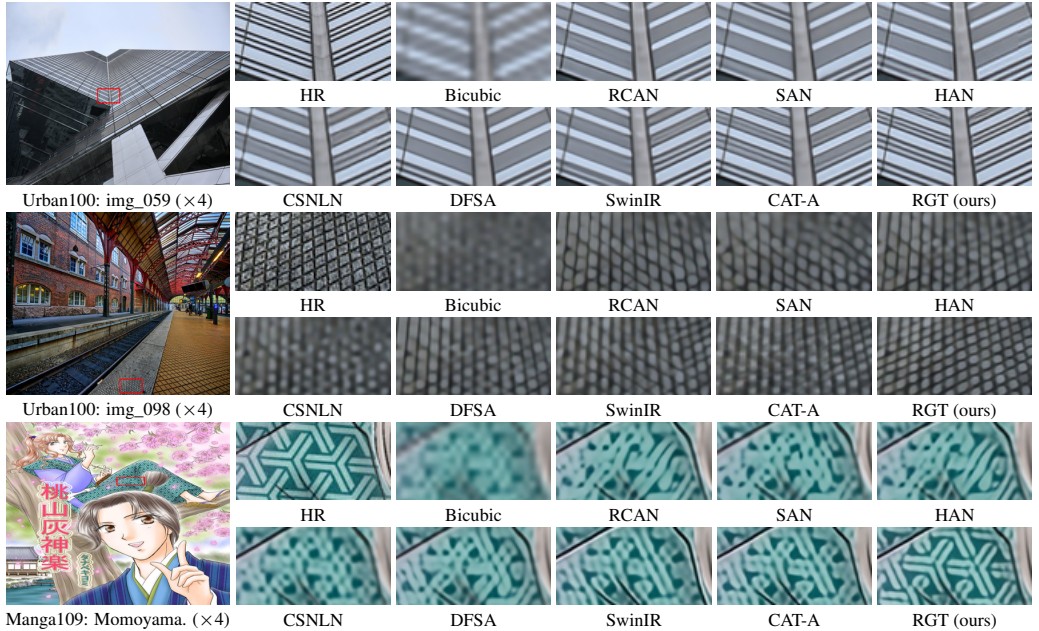

Figure 6: Visual comparison for image SR (×4) in some challenging cases.

| Method | EDSR | RCAN | HAN | CSNLN | SwinIR | CAT-A | RGT-S (ours) | RGT (ours) |
|---|---|---|---|---|---|---|---|---|
| Params(M) | 43.09 | 15.59 | 16.07 | 6.57 | 11.90 | 16.60 | 10.20 | 13.37 |
| FLOPs(G) | 823.34 | 261.01 | 269.1 | 84,155.24 | 215.32 | 360.67 | 193.08 | 251.07 |
| Urban100 | 26.64 | 26.82 | 26.85 | 27.22 | 27.45 | 27.89 | 27.89 | 27.98 |
| Manga109 | 31.02 | 31.22 | 31.42 | 31.43 | 32.03 | 32.39 | 32.38 | 32.50 |

Table 3: Comparison of parameters, FLOPs, and PSNR (dB) values on Urban100 and Manga109 with scaling factor ×4. When we calculate FLOPs, the output size is $3×512×512$.

## 4.4 MODEL SIZE ANALYSES

We further show the comparison of parameter numbers, FLOPs, and performance with recent image SR methods in Tab. 3. FLOPs are measured when the output size is set as $3×512×512$, and PSNR values are tested on Urban100 and Manga109 (×4). Our RGT has lower computational complexity and model size than CNN-based methods, EDSR (Lim et al., 2017) and RCAN (Zhang et al., 2018a). Compared with CSNLN (Mei et al., 2020), our RGT only requires 0.3% computational complexity (*i.e.*, FLOPs). Meanwhile, compared with the recent Transformer-based model, CAT-A, our RGT performs better, while FLOPs decreased by 30.39% (109.6G) and parameters decreased by 19.46% (3.23M). Compared with SwinIR (Liang et al., 2021), RGT has comparable computational complexity and model size. Furthermore, to further demonstrate the effectiveness of our method, we provide another version of the model, RGT-S, with lower FLOPs and parameters than SwinIR. Our RGT-S still obtains notable SR performance gains compared with other methods. These comparisons indicate that our method achieves a better trade-off between model complexity and performance.

## 5 CONCLUSION

We propose a new Transformer model, named Recursive Generalization Transformer (RGT), for accurate image SR. Our RGT is capable of modeling global spatial information while maintaining low computational costs. Specifically, we design the recursive-generalization self-attention (RG-SA) to extract global information effectively in linear complexity. The RG-SA computes cross-attention between the input features and the representative maps recursively aggregated from the input. Meanwhile, the channel dimensions of attention matrices are further scaled to mitigate the redundancy in the channel domain. Furthermore, to improve the exploitation of the global context, we combine RG-SA with local self-attention, and propose the hybrid adaptive integration (HAI) for module integration. The HAI acts on the outside of each Transformer block to directly fuse features at different levels (local or global). Extensive experiments on image SR demonstrate that our proposed RGT achieves superior performance over recent state-of-the-art methods.

## ACKNOWLEDGMENTS

This work was supported in part by NSFC grant (62141220, U19B2035) and Shanghai Municipal Science and Technology Major Project (2021SHZDZX0102).

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
