# SUPPLEMENTARY MATERIAL:
# RECURSIVE GENERALIZATION TRANSFORMER FOR IMAGE SUPER-RESOLUTION

**Zheng Chen**[1], **Yulun Zhang**[1*†], **Jinjin Gu**[2,3], **Linghe Kong**[1*], **Xiaokang Yang**[1]
[1]Shanghai Jiao Tong University, [2]Shanghai AI Laboratory, [3]The University of Sydney

## 1   MORE VARIANTS OF RGT

We provide more versions of RGT. **(1)** We adopt the same window attention (size: 8×8) as SwinIR (Liang et al., 2021) and provide RGT-sw. **(2)** We provide a parallel variant of our method, RGT-parallel, which applies the parallel arrangement of RG-SA and L-SA.

| Method | Params | FLOPs | Set5 | | Set14 | | B100 | | Urban100 | | Manga109 | |
|---|---|---|---|---|---|---|---|---|---|---|---|---|
| | | | PSNR | SSIM | PSNR | SSIM | PSNR | SSIM | PSNR | SSIM | PSNR | SSIM |
| SwinIR (Liang et al., 2021) | 11.75M | 205.31G | 38.42 | 0.9623 | 34.46 | 0.9250 | 32.53 | 0.9041 | 33.81 | 0.9427 | 39.92 | 0.9797 |
| RGT-sw | 11.60M | 187.73G | **38.46** | **0.9624** | **34.54** | **0.9257** | **32.54** | **0.9043** | **33.90** | **0.9433** | **40.12** | **0.9802** |

Table 1: Quantitative comparison (×2). The input size is 3×128×128 to calculate FLOPs.

### 1.1   RGT-SW: 8×8 WINDOW SIZE

Our method applies Rwin-SA as the local self-attention (L-SA) by default. To further demonstrate the effectiveness of our method, we adopt the same window attention as SwinIR (Liang et al., 2021). For a fair comparison, we set the window size to 8×8, the same as SwinIR.

**Implementation Details.** We still utilize the structure of our RGT. We set the window size in the L-SA block as 8×8. We denote the new version of RGT as RGT-sw. We set the residual group number (RG) and the Transformer block number in each RG as 7 and 6, respectively. We set the channel dimension number, multi-head number, and mlp expansion ratio as 180, 6, and 2 for each Transformer block. We set stride size $s_r$ as 4 and adjustment factor $c_r$ as 0.5 for RG-SA.

**Training Settings.** We train RGT-sw on DIV2K (Timofte et al., 2017) and Flickr2K (Lim et al., 2017), and test it on five datasets: Set5 (Bevilacqua et al., 2012), Set14 (Zeyde et al., 2010), B100 (Martin et al., 2001), Urban100 (Huang et al., 2015), and Manga109 (Matsui et al., 2017). We conduct experiments on the scale factor: ×2. The training settings are the same as for RGT-S and RGT. More details are shown in the main paper.

**Quantitative results.** We compare our RGT-sw with SwinIR (Liang et al., 2021). The results are listed in Tab. 1. When we calculate FLOPs, the input size is set as 3×128×128. Our RGT-sw outperforms SwinIR on all datasets, while the Params and FLOPs are lower than SwinIR. In particular, RGT surpassed SwinIR by 0.2 dB on the Manga109 dataset.

### 1.2   RGT-PARALLEL: PARALLEL ARRANGEMENT

We adopt the serial arrangement of RG-SA and L-SA in RGT. We also attempt the parallel arrangement. Specifically, we split the input feature into two parts evenly along the channel dimension. One part is for RG-SA, and another part is for L-SA. Then we concatenate the outputs of the two attention modules, and use the linear projection to fuse the features.

**Implementation Details.** We apply the same architecture as RGT-S to RGT-parallel, and adopt the parallel arrangement in all Transformer blocks. And the implementation details (*e.g.*, channel dimension number and window size) of RGT-parallel are the same as RGT-S.

---

*Corresponding authors: Yulun Zhang, yulun100@gmail.com; Linghe Kong, linghe.kong@sjtu.edu.cn.
†The work was mainly done when Yulun Zhang was at ETH Zurich.

| Method | Params | FLOPs | Set5 | | Set14 | | B100 | | Urban100 | | Manga109 | |
|---|---|---|---|---|---|---|---|---|---|---|---|---|
| | | | PSNR | SSIM | PSNR | SSIM | PSNR | SSIM | PSNR | SSIM | PSNR | SSIM |
| SwinIR (Liang et al., 2021) | 11.75M | 205.31G | 38.42 | 0.9623 | 34.46 | 0.9250 | 32.53 | 0.9041 | 33.81 | 0.9427 | 39.92 | 0.9797 |
| RGT-parallel | 9.03M | 183.64G | 38.55 | 0.9626 | 34.64 | 0.9259 | 32.56 | 0.9045 | 34.23 | 0.9452 | 40.13 | 0.9804 |
| RGT-S | 10.20M | 193.08G | **38.56** | **0.9627** | **34.77** | **0.9270** | **32.59** | **0.9050** | **34.32** | **0.9457** | **40.18** | **0.9805** |

Table 2: Quantitative comparison (×2). The input size is 3×128×128 to calculate FLOPs.

**Training Settings.** We train RGT-sw on DIV2K (Timofte et al., 2017) and Flickr2K (Lim et al., 2017). The training settings are consistent with RGT-S and RGT. The main paper has more details.

**Quantitative results.** We compare the RGT-parallel with SwinIR (Liang et al., 2021) and RGT-S. The results are listed in Tab. 2. (1) Our RGT-parallel outperforms SwinIR on all datasets, while the Params and FLOPs are similar or lower than SwinIR. (2) Compared with RGT-S, the performance of RGT-parallel slightly decreases. For instance, the PSNR value drops 0.09 dB on Urban100. However, the overall performance difference between RGT-S and RGT-parallel is not great. It indicates that parallel arrangement can serve as another subsequent exploration direction.

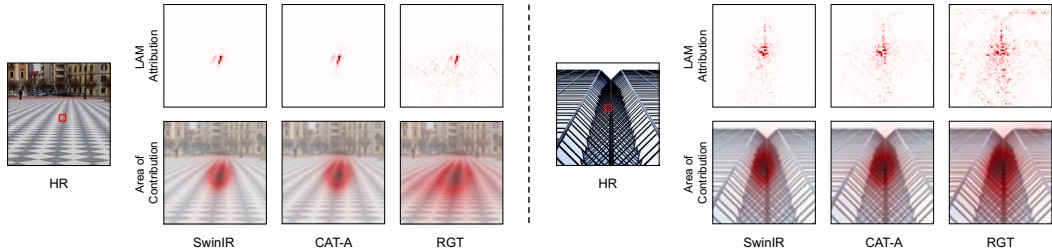

Figure 1: LAM (Gu & Dong, 2021) comparison between SwinIR (Liang et al., 2021), CAT-A (Chen et al., 2022), and RGT. LAM attribution maps represent the utilized pixels in the input LR image when reconstructing the red box region in the HR image. We also illustrate the area of contribution.

## 2 LAM RESULTS

We apply LAM (Gu & Dong, 2021) to visualize the receptive fields of SwinIR, CAT-A, and our RGT in Fig. 1. The LAM attribution maps represent the utilized pixels in the input LR image when reconstructing the red box region in the HR image. The utilized pixels are marked in red. Therefore, the more pixels are used to recover, the wider the distribution of red points is in the LAM attribution map, and the larger the receptive field of the model. And we illustrate the area of contribution.

As shown in Fig. 1, the marked pixels of our RGT in LAM attribution maps extended to almost complete images. However, the compared methods, SwinIR and CAT, only utilize a limited range of pixels. It indicates that our method can more effectively utilize global information for image SR, compared to previous local attention methods. These results are also consistent with quantitative and visual comparisons, showing the effectiveness of our method.

## 3 MORE CKA VISUALIZATIONS

In this part, we calculate centered kernel alignment (CKA) (Cortes et al., 2012; Kornblith et al., 2019; Raghu et al., 2021) similarities between all pairs of Transformer blocks in completed trained RGT-S and RGT. Meanwhile, to further analyze the effects of HAI, we directly remove the learnable adapter $\alpha$ in RGT-S (RGT), denoted as RGT-S (RGT) rm alpha. Also, we remove the HAI in RGT-S (RGT), denoted as RGT-S (RGT) rm HAI. Please note that, unlike the RGT-S w/o HAI in Fig. 4 in the main paper, for RGT-S (RGT) rm alpha and RGT-S (RGT) rm HAI, we do not train them from scratch. The results are shown as heatmaps in Fig. 2.

***First, models of different sizes exhibit similar structures.*** Comparing the CKA results of RGT-S and RGT, we find that their network structures are similar, although the number of blocks differs.

***Second, the learnable adapter promotes module integration.*** We observe that the representation similarity between different residual groups (RGs, each RG contains 6 blocks) has significant dif-

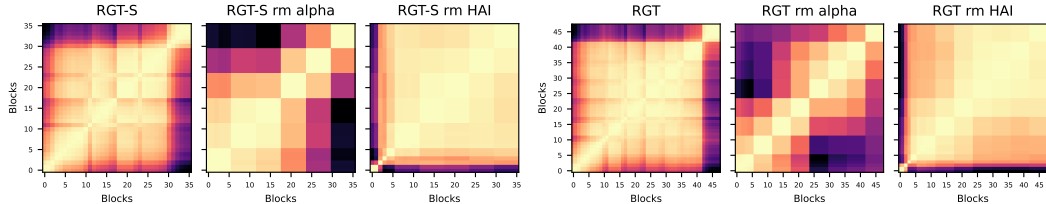

Figure 2: CKA similarity between all pairs of Transformer blocks. For RGT-S (RGT) rm alpha and RGT-S (RGT) rm HAI, unlike RGT w/o HAI in the main paper, we directly remove the learnable adapter $\alpha$ and HAI in the completed trained RGT-S (RGT). The model RGT-S w/o HAI in the main paper is constructed by removing HAI in RGT-S and trained from scratch.

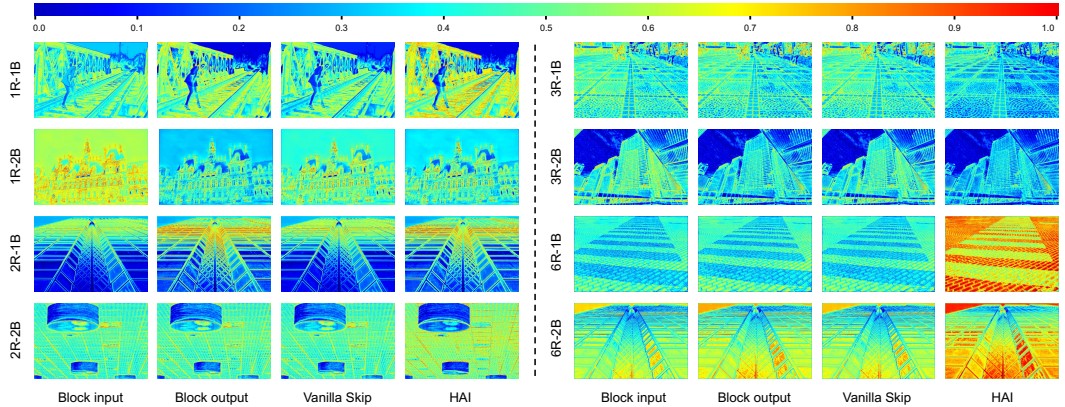

Figure 3: Visualization of features in RGT. We provide more features of Transformer blocks located at different positions in RGT. Each block is indexed as $n$R-$m$B: the $m^{th}$ Transformer block in the $n^{th}$ residual group (RG). Both the index number of the RG and the index number of the Transformer block in each RG start from 1. Please zoom in for a better view.

ferences, when removing $\alpha$ (RGT-S rm alpha and RGT rm alpha). In contrast, the CKA results between the RGs of the complete model (RGT-S and RGT) are closer.

***Third, the HAI improves the information flow.*** From RGT-S rm HAI and RGT rm HAI, we discover that the similarity between initial blocks ($0^{th}$~$2^{nd}$) and subsequent blocks is very low. On the contrary, with HAI, the information in shallow layers can effectively flow to the deep layers, thereby enhancing the similarity of blocks ($0^{th}$~$2^{nd}$) to other blocks.

## 4 MORE FEATURE VISUALIZATIONS

We further visualize feature maps of more Transformer blocks in RGT as a supplement to the visualization in the main paper. The results are shown in Fig. 3. We can discover that in the shallow block, the difference between input and output is large, which is consistent with the CKA results in Fig. 2. In addition, by directly adding input and output, feature fusion is not effective. Especially in deep layer blocks (*e.g.*, 6R-2B), when the input is similar to the output, the vanilla skip connection (He et al., 2016) has little effect. In contrast, the results of HAI change obviously compared to other features and present richer details. These results further demonstrate the effectiveness of HAI.

## 5 MORE QUANTITATIVE RESULTS

We compare our RGT-S with SwinIR (Liang et al., 2021) on high-resolution datasets: Tset2K/4K/8K (Kong et al., 2021). The results are listed in Tab. 3. When we calculate FLOPs, the input size is set as $3\times128\times128$. Our RGT-S outperforms SwinIR on all datasets with lower Params and FLOPs. For instance, RGT-S obtains 0.17 dB gains on very large images, *i.e.*, Test8K.

| Method | Params | FLOPs | Test2K | | Test4K | | Test8K | |
|--------|--------|-------|--------|--------|--------|--------|--------|--------|
| | | | PSNR | SSIM | PSNR | SSIM | PSNR | SSIM |
| SwinIR (Liang et al., 2021) | 11.75M | 205.31G | 27.99 | 0.7898 | 29.48 | 0.8349 | 35.57 | 0.9034 |
| RGT-S | 10.20M | 193.08G | **28.10** | **0.7920** | **29.60** | **0.8368** | **35.74** | **0.9051** |

Table 3: Quantitative comparison on high-resolution datasets: Tset2K/4K/8K (Kong et al., 2021).

# 6  MORE VISUAL COMPARISONS

In Figs. 5 and 6, we provide more visual comparisons on the Urban100 and Manga109 datasets. Our RGT handles several challenging cases better and recovers more details than compared methods. For instance, in img_044 of the Urban100, most compared methods fail to restore the horizontal textures, while our RGT produces much sharper details. In DualJustice of the Manga109, our RGT recovers the white lines clearly. In construct, most compared methods suffer blurring artifacts. Similar observations can be found in other cases. These results can be used as a supplement to the visual comparison of the main paper, and further demonstrate the superiority of our methods.

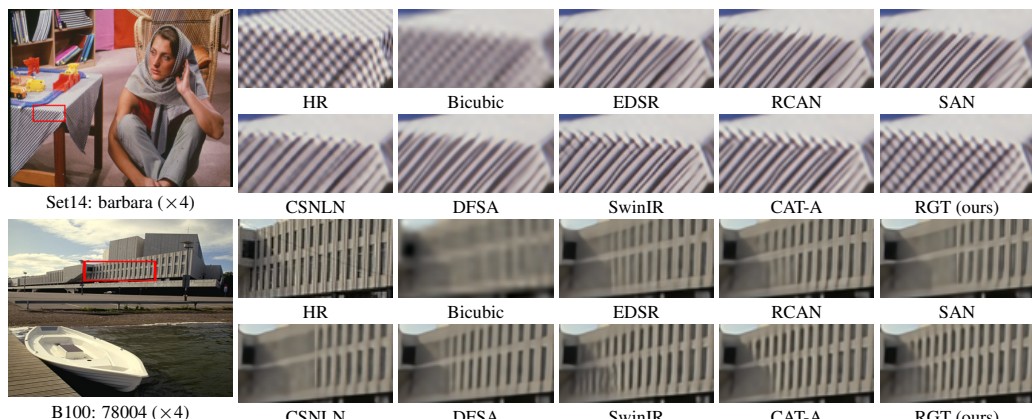

Figure 4: Visual comparison for image SR (×4) on Set14 and B100.

# 7  MORE VISUAL COMPARISONS ON SET14 AND B100

In Fig. 4, we provide more visual comparisons on the Set14 (Zeyde et al., 2010) and B100 (Martin et al., 2001) datasets. Our RGT also demonstrates superior visual results on these datasets. For instance, in barbara of the Set14, our RGT successfully recovers more distinct texture features than compared methods. These results further showcase the effectiveness of our approach.

# 8  LIMITATIONS AND FUTURE WORK

Due to the global information modeling ability, our Recursive Generalization Transformer (RGT) achieves state-of-the-art performance on image SR. However, we have not applied RGT to more image SR tasks, such as real-world image SR and blind image SR. Furthermore, for the design of the hybrid adaptive integration (HAI), we adopt the learnable adapter and skip connection. We could investigate more fusion methods in further work. For example, adjust input and output simultaneously, or utilize concatenation operation to fuse input and output.

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

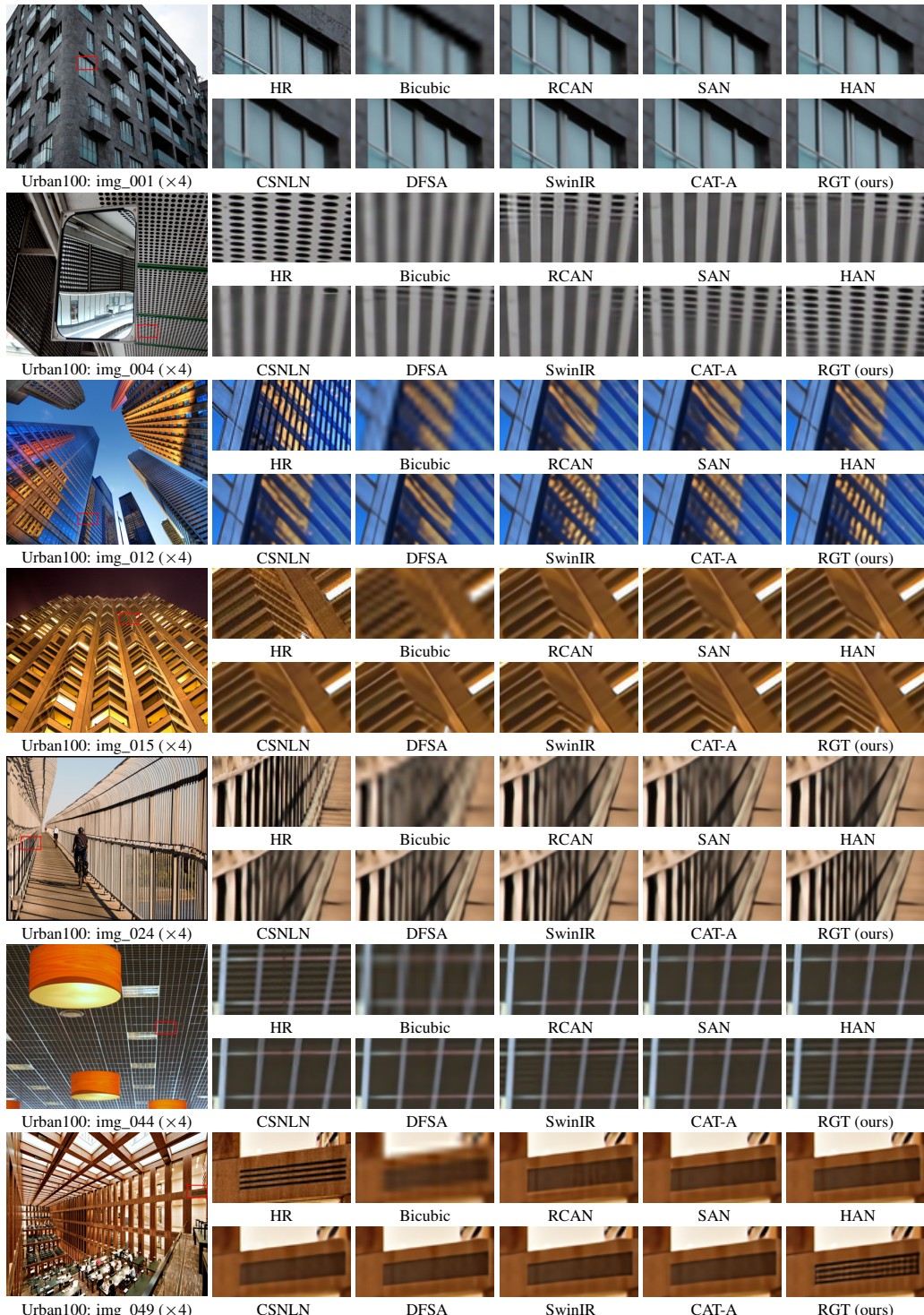

Figure 5: Visual comparison for image SR (×4) in some challenging cases.

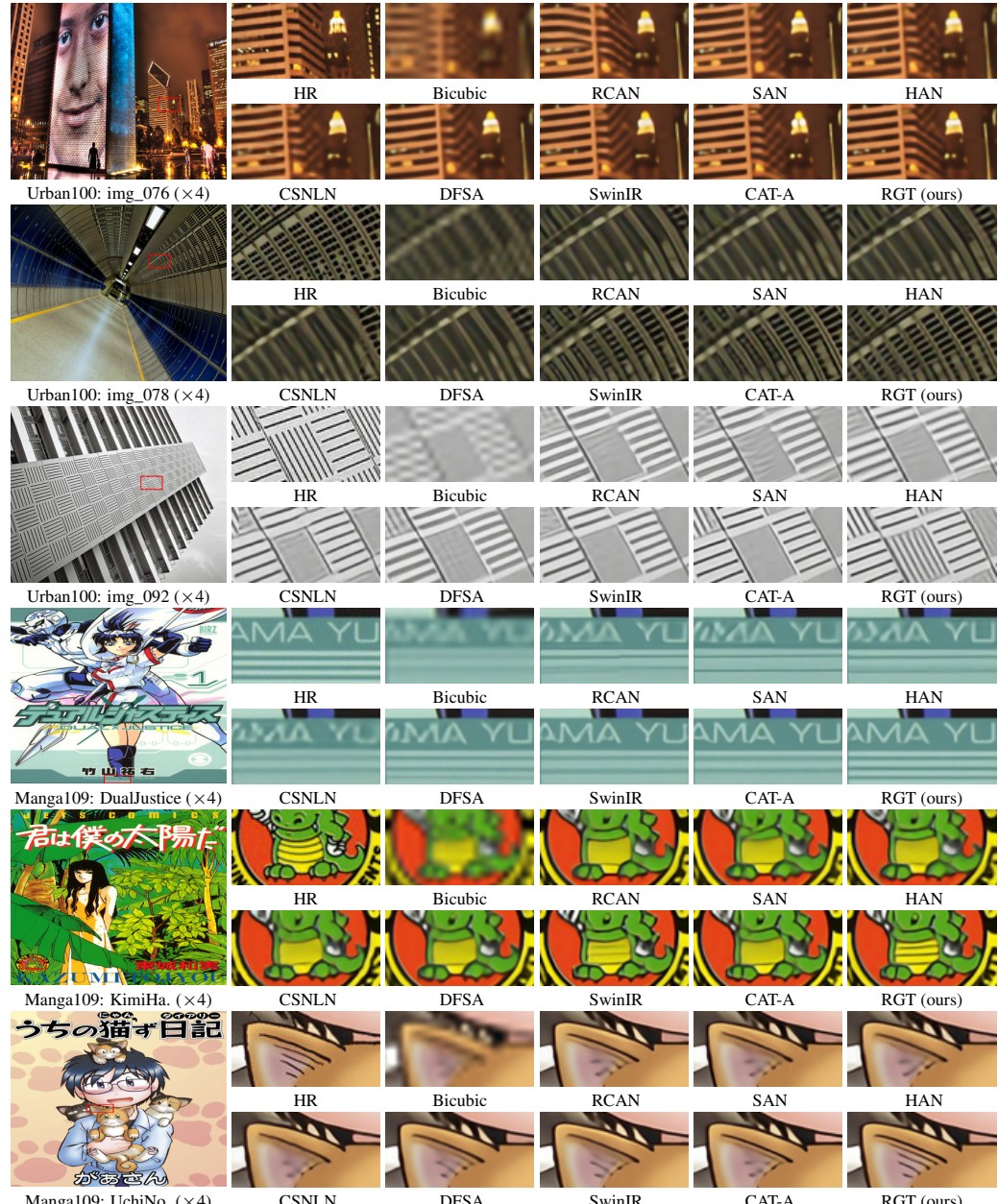

Figure 6: Visual comparison for image SR (×4) in some challenging cases.