# OpenReview forum: "Recursive Generalization Transformer for Image Super-Resolution"
_ICLR.cc/2024/Conference — ICLR 2024 poster_

### Official Review · Reviewer_bq6U · 2023-10-27

**Soundness:** 3 good
**Presentation:** 3 good
**Contribution:** 3 good
**Rating:** 8
**Confidence:** 4

**Summary:**

This paper proposes a recursive generalization transformer for image SR. In which, the recursive-generalization self-attention (RG-SA) recursively aggregates input features into representative feature maps, and then extract global information via the cross-attention. Moreover, the authors combine the RG-SA with local self-attention to enhance the exploitation of the global context, and further propose the hybrid adaptive integration (HAI) for module integration. Experiments demonstrate that the proposed RGT achieves good performance.

**Strengths:**

1.The paper is well writting and easy to following.
2.The author proves the effectiveness of the proposed components by sufficient experiments.

**Weaknesses:**

1. The comparisons with recent ViT-based methods are insufficient.
2. The author should also make comparisons on inference speed and GPU usage to evaluate the model computational efficiency.
3. What is the performance of a larger window SA with HAI?
4. In the ablation of RG-SA, the authors state that "a smaller cr reduces redundancy between channels". Why does fewer channels reduce feature redundancy?
5. What are the implementation details of dynamically choosing the recursion time T in RGM？

**Questions:**

See the weaknesses part

---

> ### Author Response · Authors · 2023-11-17
> **Response to Reviewer bq6U (denoted as R4) part 1**
>
> `Q4-1:` The comparisons with recent ViT-based methods are insufficient.
>
> `A4-1:`
>
> Thanks for pointing it out. We compare our method with more ViT-based methods: EDT [ref1], ART [ref2], and DAT [ref3].
>
> The models are tested on Urban100 (×2). The input size for calculating FLOPs is 3×256×256. The results are as follows:
>
> | Method           | Params |   FLOPs   | PSNR(dB) |  SSIM  |
> | ---------------- | :----: | :-------: | :------: | :----: |
> | EDT-B (arXiv'22) | 11.48M | 1,105.16G |  33.81   | 0.9427 |
> | ART-S (ICLR'23)  | 11.72M | 1,228.98G |  34.02   | 0.9437 |
> | DAT-S (ICCV'23)  | 11.06M |  773.35G  |  34.12   | 0.9444 |
> | RGT-S (ours)     | 10.05M |  793.45G  |  34.32   | 0.9457 |
>
> 1. Compared to EDT-B and ART-S, our method has fewer FLOPs and Params while obtaining better performance.
> 2. Compared to DAT-S, our RGT-S demonstrates superior performance (improving by 0.2dB of PSNR) with similar FLOPs and Params.
> 3. These results further demonstrate the effectiveness of the linear global attention proposed in our method for the SR task.
>
>
>
> [ref1] Wenbo Li, Xin Lu, Jiangbo Lu, Xiangyu Zhang, and Jiaya Jia. On efficient transformer and image pre-training for low-level vision. arXiv preprint arXiv:2112.10175, 2021.
>
> [ref2] Jiale Zhang, Yulun Zhang, Jinjin Gu, Yongbing Zhang, Linghe Kong, XinYuan. Accurate Image Restoration with Attention Retractable Transformer, In ICLR, 2023.
>
> [ref3] Zheng Chen, Yulun Zhang, Jinjin Gu, et al. Dual Aggregation Transformer for Image Super-Resolution, In ICCV, 2023.
>
>
>
> `Q4-2:` The author should also make comparisons on inference speed and GPU usage to evaluate the model computational efficiency.
>
> `A4-2:`
>
> Thanks for your valuable suggestions. We compare SwinIR, CAT-A, RGT-S, and RGT. The tests are conducted on one A100 GPU at scale ×4. We evaluate both inference speed (latency) and GPU usage. The input size is set to 3×128×128. For accuracy, we calculated the average time with 100 inputs.
>
> | Method                | SwinIR | CAT-A  | RGT-S  |  RGT   |
> | --------------------- | :----: | :----: | :----: | :----: |
> | Inference Speed (/ms) | 121.67 | 240.10 | 116.19 | 157.77 |
> | GPU usgae (GiB)       | 18.78  | 19.50  | 19.54  | 22.97  |
>
> 1. For **inference speed**, our RGT-S is faster than SwinIR and CAT-A. Meanwhile, RGT's speed is superior to CAT-A. This demonstrates the effectiveness of our approach.
> 2. For **GPU usage**, our method consumes less memory than CAT-A. Compared to SwinIR, the memory usage is slightly higher, which may be due to implementation aspects of the code. For example, using `contiguous()` operations in PyTorch can consume more memory to speed up computation. We plan to optimize the code implementation to reduce GPU usage.

---

> ### Author Response · Authors · 2023-11-17
> **Response to Reviewer bq6U (denoted as R4) part 2**
>
> `Q4-3:` What is the performance of a larger window SA with HAI?
>
> `A4-3:`
>
> Thanks for your question. We conduct experiments to compare different window sizes in window SA.
>
> We compared the 8×32 window (used in our RGT-S) with the 32×32 window. Other settings of the two models are consistent with RGT-S. Both models are trained under the same settings with iterations=50K. We test models on Urban100 (x2). The input size for calculating FLOPs is 3×256×256.
>
> | Method | Params(M) | FLOPs(G) | PSNR(dB) |  SSIM  |
> | ------ | :-------: | :------: | :------: | :----: |
> | 8×32   |   10.05   |  793.45  |  32.23   | 0.9301 |
> | 32×32  |   10.05   | 1,119.61 |  32.32   | 0.9306 |
>
> 1. Increasing the window size can improve the model's performance (increase PSNR by 0.09dB).
> 2. However, this also increases the computational complexity (increase FLOPs by 326.16 G).
> 3. Considering both performance and computational cost, we select the current 8×32 window setting.
>
>
>
> `Q4-4:` In the ablation of RG-SA, the authors state that "a smaller cr reduces redundancy between channels". Why does fewer channels reduce feature redundancy?
>
> `A4-4:`
>
> Thank you for your questions. We explain below:
>
> 1. **For redundancy:** Since we aggregate feature maps through the recursive generalization module, maintaining the original number of channels could lead to a certain level of redundancy.
>
> 2. **For reducing redundancy:** By scaling down the channels, we reduce the number of feature maps. This helps to enhance information aggregation and increase sensitivity (discrimination) between different feature maps. Consequently, when computing cross-attention, this allows for better focus on essential features and improves information transfer.
>
> 3. **Experiments:** We compared models with different numbers of channels in our ablation study, with results as follows. The input size for calculating FLOPs was 3×128×128.
>
>    | Method    | Params(M) | FLOPs(G) | PSNR(dB) |  SSIM  |
>    | --------- | :-------: | :------: | :------: | :----: |
>    | $c_r$=1   |   11.37   |  189.62  |  33.54   | 0.9404 |
>    | $c_r$=0.5 |   10.05   |  183.08  |  33.68   | 0.9414 |
>
>    Reducing the number of channels effectively improves model performance, which further supports our analyses.
>
>
>
> `Q4-5:` What are the implementation details of dynamically choosing the recursion time T in RGM？
>
> `A4-5:`
>
> Thank you for your questions.
>
> The recursion time $T$ is determined by the formula $T=\lfloor\log_{s_r}{\frac{H}{h}}\rfloor$. The stride size $s_r$ is set to 4, and the representative map size $h$ is 4 for training and 16 for testing.
>
> **For $s_r=4$**, we choose this setting following the default selection in previous works [ref4].
>
> **For $h=4/16$**, we choose this to balance performance and computational complexity.
>
> 1. For training, setting $h=4$ can speed up the process (compared to h=16). Meanwhile, given the input patch size of 64×64, $h=4$ cannot lead to significant information loss and performance drop.
>
> 2. For testing, $h=4$ is too small compared to the full image resolution and would result in substantial information loss. For instance, img001.png in Urban100 (×2) is 512×322 ($\gg$4×4). Conversely, h=16 effectively improves performance without significantly increasing computational complexity.
>
> 3. We compare h=4/16/64 in RGT-S on Urban100 (×2). The input size for calculating FLOPs is 3×256×256.
>
>    | Method | Params(M) | FLOPs(G)  | PSNR(dB) |  SSIM  |
>    | ------ | :-------: | :-------: | :------: | :----: |
>    | h=4    |   10.05   |  716.82G  |  32.78   | 0.9336 |
>    | h=16   |   10.05   |  793.45G  |  34.32   | 0.9457 |
>    | h=64   |   10.05   | 2,019.55G |   N/A    |  N/A   |
>
>    **Note:** When h=64, the complexity is too high that it cannot be tested on the full image.
>
>    It is observed that compared to h=4 or 64, h=16 achieves a better balance between performance and complexity.
>
> 4. Therefore, we use h=4 for training and h=16 for testing.
>
>
>
> [ref4] Wenhai Wang, Enze Xie, Xiang Li, Deng-Ping Fan, Kaitao Song, Ding Liang, Tong Lu, Ping Luo, and Ling Shao. Pyramid vision transformer: A versatile backbone for dense prediction without convolutions. In ICCV, 2021.

---

### Official Review · Reviewer_Pj4p · 2023-10-27

**Soundness:** 4 excellent
**Presentation:** 4 excellent
**Contribution:** 4 excellent
**Rating:** 8
**Confidence:** 5

**Summary:**

This paper proposed a novel image SR method, Recursive Generalization Transformer (RGT). The RGT uses recursive-generalization self-attention (RG-SA) and hybrid adaptive integration (HAI) to model global information. Extensive experiments demonstrate that the proposed RGT achieves state-of-the-art performance on image SR.

**Strengths:**

- The paper's writing and organization are good. All illustrations, tables, and visual results are intuitive and clear.
- The motivation for the proposed method is reasonable. The global information in SR is important while reducing the complexity of global attention is crucial for its application in SR tasks.
- The proposed components RG-SA and HAI are novel and valuable.
- The ablation study is extensive. The effectiveness of each part in RGT is demonstrated.
- The authors provide multiple model versions and compare them with state-of-the-art methods. All results prove that the RGT proposed in this paper is a promising SR method.

**Weaknesses:**

- When compared with CAT-A, the improvements of RGT on some datasets (Set5, Set14) are not very obvious (< 0.1 dB).
- Although FLOPs are provided in Sec. 4.4, the running time of the model on real devices should also be provided.
- The primary evaluation metrics used in the paper are PSNR and SSIM. However, these metrics may not reflect actual SR performance. Some perceptual metrics, such as LPIPS, should be evaluated.

**Questions:**

- The author mentioned in the paper that the proposed RG-SA realizes linear complexity, which is better than other global methods. So, has the author tried other global attention, such as the attention proposed in PVT or XCiT? How do the computational complexity and performance of RG-SA compare with these methods?

---

> ### Author Response · Authors · 2023-11-17
> **Response to Reviewer Pj4p (denoted as R3)**
>
> `Q3-1:` When compared with CAT-A, the improvements of RGT on some datasets (Set5, Set14) are not very obvious (< 0.1 dB).
>
> `A3-1:`
>
> Thank you for pointing this out. We provide more analysis on this results.
>
> 1. **Fewer Params and FLOPs.** Compared to CAT-A, our RGT-S has a lower parameter count and computational complexity. Therefore, any performance improvement itself demonstrates the effectiveness of our proposed method.
>
> 2. **Global Attention.** The core of our proposed RGT is RG-SA, which effectively models global information. This is more advantageous for processing **high-resolution** images. The images in Set5 and Set14 are relatively small (e.g., bird is 144×144 in Set5-×2), hence the marginal differences. However, the improvements are more significant on high-resolution datasets (e.g., img001 is 512×322 in Urban100-×2) like Urban100 and Manga109.
>
> 3. Our comparisons at scales ×2, ×3, and ×4 between RGT and CAT-A show increasing performance improvements as we move from scale 4 to 2 (with gradually increasing input image resolution). This further validates our analysis.
>
>    | Method |  ×2   |  ×3   |  ×4   |
>    | ------ | :---: | :---: | :---: |
>    | CAT-A  | 34.26 | 30.12 | 27.89 |
>    | RGT    | 34.47 | 30.28 | 27.98 |
>    | △      | +0.21 | +0.16 | +0.09 |
>
> 4. **Visual Comparison.** Our method shows more noticeable improvements in visual comparisons. We have provided corresponding results in **Sec. 7 of the supplementary material**.
>
>
>
> `Q3-2:` Although FLOPs are provided in Sec. 4.4, the running time of the model on real devices should also be provided.
>
> `A3-2:`
>
> Thank you for your suggestions. We assess the running times at ×4 scale on one A100 GPU. The input size is 3×128×128. Our comparison includes SwinIR, CAT-A, RGT-S, and RGT. To ensure accuracy, the average time is calculated over 100 input images.
>
> | Method            | SwinIR | CAT-A  | RGT-S  |  RGT   |
> | ----------------- | :----: | :----: | :----: | :----: |
> | Running Time (ms) | 121.67 | 240.10 | 116.19 | 157.77 |
>
> 1. Our RGT-S and RGT models demonstrate faster running times compared to CAT-A.
> 2. Additionally, the running time of our RGT-S is also lower than that of SwinIR.
>
>
>
> `Q3-3:` The primary evaluation metrics used in the paper are PSNR and SSIM. However, these metrics may not reflect actual SR performance. Some perceptual metrics, such as LPIPS, should be evaluated.
>
> `A3-3:`
>
> Thank you for your suggestions. We compare our RGT with SwinIR and CAT-A using perceptual metrics LPIPS and DISTS. The test is conducted on the Manga109 and Urban100 datasets at ×4 scale.
>
> | Method | Urban100 |        | Manga109 |        |
> | ------ | :------: | :----: | :------: | :----: |
> |        |  LPIPS   | DISTS  |  LPIPS   | DISTS  |
> | SwinIR |  0.1840  | 0.1533 |  0.0926  | 0.0766 |
> | CAT-A  |  0.1801  | 0.1502 |  0.0906  | 0.0753 |
> | RGT    |  0.1759  | 0.1469 |  0.0893  | 0.0719 |
>
> Our method achieves better perceptual scores than other methods, consistent with the visual comparisons. This further demonstrates the effectiveness of our proposed method.
>
>
>
> `Q3-4:` The author mentioned in the paper that the proposed RG-SA realizes linear complexity, which is better than other global methods. So, has the author tried other global attention, such as the attention proposed in PVT or XCiT? How do the computational complexity and performance of RG-SA compare with these methods?
>
> `A3-4:`
>
> Thank you for your question. We compare our RG-SA with XCA, as proposed in XCiT, and the sparse attention proposed in ART.
>
> **XCA**: A global attention mechanism with linear complexity using a “transposed” operation, but it cannot explicitly model the spatial relationship.
>
> 1. We compare RG-SA with XCA. For a fair comparison, we directly replace RG-SA in RGT-S with XCA and get the model RGT-XCA. We train both RGT-S and RGT-XCA under the same settings (iterations: 50K). Models are tested on Manga109, with an input size of 3×256×256 for calculating FLOPs.
>
>    | Method  | Params(M) | FLOPs(G) | PSNR(dB) |  SSIM  |
>    | ------- | :-------: | :------: | :------: | :----: |
>    | RGT-XCA |   10.88   |  824.70  | 38.0601  | 0.9770 |
>    | RGT-S   |   10.05   |  793.45  | 38.4279  | 0.9771 |
>
> 2. Our RGT-S outperforms RGT-XCA while having a similar computational complexity and parameters. This demonstrates the effectiveness of RG-SA in modeling global spatial information.
>
>
>
> **Sparse attention**: A spatial global attention with quadratic complexity.
>
> 1. Due to computational resource limitations, we directly compare RGT-S with ART-S without retraining.
>
>    | Method | Params(M) | FLOPs(G) | PSNR(dB) |  SSIM  |
>    | ------ | :-------: | :------: | :------: | :----: |
>    | ART-S  |   11.72   | 1,228.98 |  40.11   | 0.9804 |
>    | RGT-S  |   10.05   |  793.45  |  40.18   | 0.9805 |
>
> 2. Our RGT-S achieves better performance with lower computational complexity. This indicates that RG-SA can effectively capture global attention while maintaining a lower (linear) computational complexity.

---

> > ### Comment · Reviewer_Pj4p · 2023-11-22
> >
> > All my concerns have been well addressed, and I have no further comments.

---

> > > ### Author Response · Authors · 2023-11-23
> > >
> > > Dear Reviewer Pj4p,
> > >
> > > Thank you for your response. We are delighted to see that our answers were able to address your concerns.
> > >
> > > Best,
> > >
> > > Authors

---

### Official Review · Reviewer_fY4p · 2023-10-30

**Soundness:** 3 good
**Presentation:** 3 good
**Contribution:** 3 good
**Rating:** 6
**Confidence:** 5

**Summary:**

This paper presents the Recursive Generalization Transformer (RGT) for image SR. Specifically, the recursive-generalization self-attention (RG-SA) is proposed to capture global spatial information. And the hybrid adaptive integration (HAI) is designed for module integration. Experiments show that the proposed RGT achieves improvements over recent methods.

**Strengths:**

1. The authors propose the recursive-generalization self-attention (RG-SA), which controls computational complexity while achieving global modeling.
2. They also design the hybrid adaptive integration (HAI). It is a simple yet effective.
3. The paper's experiments are comprehensive. The ablation study demonstrates the effects of each component.
4. Quantitative and qualitative results indicate that the proposed method outperforms SwinIR and CAT-A.
5. The authors provide various visual results: feature maps and CKA heatmap, which greatly support the author's claims and enrich the overall quality of the paper.
6. The paper is well-organized, with clear and readable content.

**Weaknesses:**

1. Some details in the paper are not clear. For example, the representative map size "h" is set as 4 for training but 16 for testing. Why not use the same settings?
2. The RGT-S and RGT all adopt a larger window size than SwinIR. To establish a fairer comparison, it is recommended to use the same window size.
3. It would be beneficial to include comparisons with more recent methods, such as RGT, to evaluate the effectiveness of the proposed method.
4. A comparison of running times should be given, which is important for practical applications of the model.

**Questions:**

1. Some details in the paper are not clear. For example, the representative map size "h" is set as 4 for training but 16 for testing. Why not use the same settings?
2. The RGT-S and RGT all adopt a larger window size than SwinIR. To establish a fairer comparison, it is recommended to use the same window size.
3. It would be beneficial to include comparisons with more recent methods, such as RGT, to evaluate the effectiveness of the proposed method.
4. A comparison of running times should be given, which is important for practical applications of the model.

**Details Of Ethics Concerns:**

M/A

---

> ### Author Response · Authors · 2023-11-17
> **Response to Reviewer fY4p (denoted as R2)**
>
> `Q2-1:` Some details in the paper are not clear. For example, the representative map size "h" is set as 4 for training but 16 for testing. Why not use the same settings?
>
> `A2-1:`
>
> Thank you for your question. We use different settings as a trade-off between complexity and performance.
>
>
>
> 1. For **training**, we set h=4 to speed up the training process (compared to h=16). Since the training input patch size is 64×64, this setting does not result in significant information loss.
>
> 2. For **testing**, we need to process the whole image, where the input image resolution may be large. For instance, the img001 in Urban100 (×2) is 512×322, much larger than 4×4. The h=4 may lose too much information, leading to a substantial decrease in performance. Therefore, we apply h=16.
>
> 3. We conduct tests with h=4, 16, and 64 in RGT-S on Urban100 (×2). The input size is 3×256×256 for calculating FLOPs.
>
>    | Method | Params(M) | FLOPs(G)  | PSNR(dB) |  SSIM  |
>    | ------ | :-------: | :-------: | :------: | :----: |
>    | h=4    |   10.05   |  716.82G  |  32.78   | 0.9336 |
>    | h=16   |   10.05   |  793.45G  |  34.32   | 0.9457 |
>    | h=64   |   10.05   | 2,019.55G |   N/A    |  N/A   |
>
>    **Note:** When h=64, the complexity is too high that it cannot be tested on the full image.
>
> 4. Compared to h=4, h=16 achieves a slight increase in complexity but a significant performance improvement. However, further increasing h would result in excessively high complexity, rendering the model inoperable.
>
> 5. In summary, we set h=4 during training and h=16 for testing to balance performance and complexity.
>
>
>
> `Q2-2:` The RGT-S and RGT all adopt a larger window size than SwinIR. To establish a fairer comparison, it is recommended to use the same window size.
>
> `A2-2:`
>
> Thank you for your valuable suggestions. We have provided a model with the same window size as SwinIR, named RGT-sw, in the supplementary material. Details in Sec. 1.1. Here, we present the corresponding table (only PSNR).
>
> | Method | Params |  FLOPs  | Set5  | Set14 | B100  | Urban100 | Manga109 |
> | ------ | :----: | :-----: | :---: | :---: | :---: | :------: | :------: |
> | SwinIR | 11.75M | 205.31G | 38.42 | 34.46 | 32.53 |  33.81   |  39.92   |
> | RGT-sw | 11.60M | 187.73G | 38.46 | 34.54 | 32.54 |  33.90   |  40.12   |
>
> Our RGT-sw, with similar computational complexity and parameters, outperforms SwinIR in performance. In particular, our RGT-sw achieves an improvement of 0.2dB on Manga109. This demonstrates the effectiveness of our proposed method.
>
>
>
> `Q2-3:` It would be beneficial to include comparisons with more recent methods, to evaluate the effectiveness of the proposed method.
>
> `A2-3:`
>
> Thank you for your suggestions. We have conducted comparisons with more recent SR methods, including EDT [ref1], ART [ref2], and DAT [ref3]. Models are tested on Urban100 (×2). The input size for calculating FLOPs is 3×256×256.
>
> | Method           | Params |   FLOPs   | PSNR(dB) |  SSIM  |
> | ---------------- | :----: | :-------: | :------: | :----: |
> | EDT-B (arXiv'22) | 11.48M | 1,105.16G |  33.81   | 0.9427 |
> | ART-S (ICLR'23)  | 11.72M | 1,228.98G |  34.02   | 0.9437 |
> | DAT-S (ICCV'23)  | 11.06M |  773.35G  |  34.12   | 0.9444 |
> | RGT-S (ours)     | 10.05M |  793.45G  |  34.32   | 0.9457 |
>
> 1. Compared with **EDR-B** and **ART-S**, our RGT-S performs better with lower computational complexity and fewer parameters.
> 2. Compared with **DAT-S**, our method shows an improvement of 0.2dB on PSNR with comparable complexity and parameters.
> 3. These comparisons underscore the effectiveness of our proposed method, which utilizes a global attention mechanism with linear complexity.
>
>
>
> [ref1] Wenbo Li, Xin Lu, Jiangbo Lu, Xiangyu Zhang, and Jiaya Jia. On efficient transformer and image pre-training for low-level vision. arXiv preprint arXiv:2112.10175, 2021.
>
> [ref2] Jiale Zhang, Yulun Zhang, Jinjin Gu, Yongbing Zhang, Linghe Kong, XinYuan. Accurate Image Restoration with Attention Retractable Transformer, In ICLR, 2023.
>
> [ref3] Zheng Chen, Yulun Zhang, Jinjin Gu, et al. Dual Aggregation Transformer for Image Super-Resolution, In ICCV, 2023.
>
>
>
> `Q2-4:` A comparison of running times should be given, which is important for practical applications of the model.
>
> `A2-4:`
>
> Thank you for your suggestions. We calculate the running times on one A100 GPU at ×4 scale. The input size is set to 3×128×128. We compare SwinIR, CAT-A, RGT-S, and RGT. For accuracy, we calculate the average time with 100 inputs.
>
> | Method            | SwinIR | CAT-A  | RGT-S  |  RGT   |
> | ----------------- | :----: | :----: | :----: | :----: |
> | Running Time (ms) | 121.67 | 240.10 | 116.19 | 157.77 |
>
> 1. Our RGT-S and RGT achieve lower running times than CAT-A.
> 2. Our RGT-S also exhibited a lower running time compared to SwinIR.

---

### Official Review · Reviewer_B1Rr · 2023-10-30

**Soundness:** 3 good
**Presentation:** 3 good
**Contribution:** 3 good
**Rating:** 8
**Confidence:** 5

**Summary:**

The authors propose the Recursive Generalization Transformer (RGT) for image super-resolution. The network can capture global spatial information and is more suitable for high-resolution images. It introduces Recursive-generalization self-attention (RG-SA) to recursively aggregate input features and Hybrid Adaptive Integration (HAI) to integrate global and local modules. Experiments show that the proposed method outperforms recent methods quantitatively and qualitatively.

**Strengths:**

1. The idea of using global attention in the Transformer is widespread. But, the authors effectively maintain low computational complexity, which is meaningful in image SR.
2. Additionally, the proposed HAI is simple yet effective. Both the ablation experiments (Table 1 (c), (d)) and the visual results (Figs. 3, 4, 5) strongly support the authors' claim: integrate global and local modules.
3. The main comparisons with recent methods demonstrate the superiority of this method. I also notice that the authors provide model size comparisons and a more fair comparison with SwinIR in the supplementary materials.
4. The paper is carefully organized. The paper is well-written and easy to read. The proposed methods are easy to follow.

**Weaknesses:**

1. The experiments on RG-SA are not enough. The authors claim the superiority of RG-SA, but it is not compared with other global attention mechanisms.
2. The authors only provide visual comparisons on Urban100 and Manga109 datasets. Comparisons on other datasets are lacking.

**Questions:**

1. The authors should compare RG-SA with other global attention, such as sparse attention proposed in ART (Accurate Image Restoration with Attention Retractable Transformer).
2. It is suggested to add more comparisons with more SR methods like ART.
3. Providing visual comparison results on more datasets to strengthen the effectiveness of the model.
4. Compared with other Transformer methods like SwinIR, are their training datasets the same? Are there any differences in the training strategy?

---

> ### Author Response · Authors · 2023-11-17
> **Response to Reviewer B1Rr (denoted as R1) part 1**
>
> `Q1-1:` The experiments on RG-SA are not enough. The authors claim the superiority of RG-SA, but it is not compared with other global attention mechanisms.
>
> `A1-1:`
>
> Thank you for your valuable feedback. We conduct additional experiments to compare RG-SA with other global attention mechanisms.
>
>
>
> 1. We compare RG-SA with the **cross-covariance attention** (**XCA**) proposed in XCiT [ref1]. XCA is a global attention mechanism with linear complexity, implemented using a "transposed" approach.
>
>    For a fair comparison, we replace RG-SA in RGT-S with XCA to build the model RGT-XCA. Both RGT-S and RGT-XCA are trained under the same settings (iterations: 50K). Models are tested on Manga109 (×2), with an input size of 3×256×256 for calculating FLOPs.
>
>    | Method  | Params(M) | FLOPs(G) | PSNR(dB) |  SSIM  |
>    | ------- | :-------: | :------: | :------: | :----: |
>    | RGT-XCA |   10.88   |  824.70  | 38.0601  | 0.9770 |
>    | RGT-S   |   10.05   |  793.45  | 38.4279  | 0.9771 |
>
>    Our RGT-S outperforms RGT-XCA, while having a similar computational complexity and parameters. This demonstrates the effectiveness of RG-SA in modeling global **spatial** information.
>
>
>
> 2. We compared RG-SA with the **sparse attention** proposed in ART [ref2], which is a global spatial attention with quadratic complexity.
>
>    Considering that ART also uses a structure combining global and local window attention, similar to our RGT, we directly compare RGT-S and ART-S without retraining, due to limited computational resources.
>
>    Note that the training settings of ART and our RGT are the same. Therefore, the comparison is fair. Models are tested on Manga109 (×2), with an input size of 3×256×256 for calculating FLOPs.
>
>    | Method | Params(M) | FLOPs(G) | PSNR(dB) |  SSIM  |
>    | ------ | :-------: | :------: | :------: | :----: |
>    | ART-S  |   11.72   | 1,228.98 |  40.11   | 0.9804 |
>    | RGT-S  |   10.05   |  793.45  |  40.18   | 0.9805 |
>
>    Our RGT-S achieves superior performance with much lower computational complexity. This highlights RG-SA's ability to effectively capture global attention while maintaining a lower (linear) computational complexity.
>
>
>
> **In conclusion**, comparisons with other global attention mechanisms demonstrate the superiority of our proposed RG-SA.
>
>
>
> [ref1] Alaaeldin Ali, Hugo Touvron, Mathilde Caron, Piotr Bo- janowski, Matthijs Douze, et al. Xcit: Cross-covariance image transformers, In NeurIPS, 2021.
>
> [ref2] Jiale Zhang, Yulun Zhang, Jinjin Gu, Yongbing Zhang, Linghe Kong, XinYuan. Accurate Image Restoration with Attention Retractable Transformer, In ICLR, 2023.

---

> ### Author Response · Authors · 2023-11-17
> **Response to Reviewer B1Rr (denoted as R1) part 2**
>
> `A1-2:` The authors only provide visual comparisons on Urban100 and Manga109 datasets. Comparisons on other datasets are lacking.
>
> `Q1-2:`
>
> Thanks for pointing it out. We have added additional visual comparisons on the Set14 and B100 datasets in **Sec.7 of the supplementary material**.
>
>
>
> `A1-3:` The authors should compare RG-SA with other global attention, such as sparse attention proposed in ART (Accurate Image Restoration with Attention Retractable Transformer).
>
> `Q1-3:`
>
> Thanks for your valuable suggestions. We compare RG-SA and other global attention mechanisms, such as XCA from XCiT and sparse attention from ART. Our experiments demonstrate the effectiveness of our proposed RG-SA.
>
>
>
> **We have responded to another similar question, `Q1-1`. Please refer to `A1-1` for more details.**
>
>
>
> `Q1-4:` It is suggested to add more comparisons with more SR methods like ART.
>
> `A1-4:`
>
> Thank you for your valuable suggestions. We conduct comparisons with more SR methods, including EDT [ref3], ART [ref2], and DAT [ref4]. Models are tested on Urban100 at scale ×2. The input size used for calculating FLOPs is 3×256×256.
>
> | Method           | Params |   FLOPs   | PSNR(dB) |  SSIM  |
> | ---------------- | :----: | :-------: | :------: | :----: |
> | EDT-B (arXiv'22) | 11.48M | 1,105.16G |  33.81   | 0.9427 |
> | ART-S (ICLR'23)  | 11.72M | 1,228.98G |  34.02   | 0.9437 |
> | DAT-S (ICCV'23)  | 11.06M |  773.35G  |  34.12   | 0.9444 |
> | RGT-S (ours)     | 10.05M |  793.45G  |  34.32   | 0.9457 |
>
> 1. Compared with **EDT-S** and **ART-S**, our method performs better with lower computational complexity and fewer parameters.
> 2. Compared with **DAT-S**, our RGT-S has similar computational complexity and parameters. However, our RGT-S exhibits an improvement of 0.2dB on PSNR.
> 3. These comparisons collectively demonstrate the effectiveness of our proposed method.
>
>
>
> [ref3] Wenbo Li, Xin Lu, Jiangbo Lu, Xiangyu Zhang, and Jiaya Jia. On efficient transformer and image pre-training for low-level vision. arXiv preprint arXiv:2112.10175, 2021.
>
> [ref4] Zheng Chen, Yulun Zhang, Jinjin Gu, et al. Dual Aggregation Transformer for Image Super-Resolution, In ICCV, 2023.
>
>
>
> `Q1-5:` Providing visual comparison results on more datasets to strengthen the effectiveness of the model.
>
> `A1-5:`
>
> Thank you for your suggestions. We have provided visual comparisons on more datasets  in **Sec.7 of the supplementary material**. We will further provide more visual comparisons and ensure timely updates.
>
>
>
> `Q1-6:` Compared with other Transformer methods like SwinIR, are their training datasets the same? Are there any differences in the training strategy?
>
> `A1-6:`
>
> Thank you for your question. The training of RGT is entirely consistent with that of SwinIR, including the use of the same training datasets (DIV2K and Flickr2K) and the same training settings.

---

### Author Response · Authors · 2023-11-17
**Response to all reviewers and area chairs for a brief summary**

Dear reviewers and area chairs,

We sincerely thank all reviewers and area chairs for their valuable time and insightful comments.



We are pleased to note that:

1. Reviewer B1Rr finds our design meaningful, and Reviewer Pj4p considers our ideas novel.
2. Reviewers B1Rr, fY4p, and Pj4p acknowledge that our method achieves state-of-the-art performance.
3. All reviewers appreciate the well-written manuscript and the extensive of our experiments.



We have responded to each reviewer individually and would like to summarize our responses here:

1. We conduct experiments comparing our proposed RG-SA with **other global attentions**: XCA and Sparse Attention, demonstrating the effectiveness of our proposed **linear complexity global spatial attention**.
2. We provide comparisons with more **Transformer-based** SR methods.
3. We provide additional **visual comparisons** on more datasets, and quantitative comparisons on more (**perceptual**) metrics.
4. We clarify details in the paper, including **training settings**, the choice of **the representative map size** $h$, and the implementation of **the recursion time** $T$.
5. We analyze the **reasons** for improvements on some datasets that are not obvious.
6. We provide comparisons of **running times** and **GPU usage**.
7. We experiment with **different window sizes** to evaluate their impact on performance.
8. We clarify why a smaller $c_r$ **reduces redundancy**.

We extend our gratitude again to all reviewers and area chairs!

Best regards,

Authors

---

### Meta-Review · Program_Chairs · 2023-12-11

**Metareview:**

This paper proposes a recursive generalization Transformer to solve image super-resolution. All reviewers are in favor of this paper. Based on the recommendations of reviewers, the paper is accepted.

**Justification For Why Not Higher Score:**

Although all reviewers are in favor of this manuscript, Reviewer fY4p concerns about the different settings about the representative map size in training and test stages. The authors provide experimental results to solve the concerns of the reviewers. However, there still exists concerns. For example, what are the results if h=16 is used in both training and test stages.

Moreover, the running time of the proposed are not significantly better than SwinIR.

**Justification For Why Not Lower Score:**

All reviewers appreciate the contributions of the paper. The minimum rating of reviewers is 6 (marginally above the acceptance threshold).

---

### Decision · Program_Chairs · 2024-01-16

Accept (poster)